# Distribution-Aligned Decoding for Efficient LLM Task Adaptation

**Senkang Hu**[1,2,*], **Xudong Han**[3,*], **Jinqi Jiang**[4], **Yihang Tao**[1,2], **Zihan Fang**[1,2]
**Yong Dai**[5], **Sam Tak Wu Kwong**[6], **Yuguang Fang**[1,2,†]
[1]Hong Kong JC STEM Lab of Smart City, [2]City University of Hong Kong,
[3]University of Sussex, [4]Huazhong University of Science and Technology,
[5]Fudan University, [6]Lingnan University
{senkang.forest, yihang.tommy, zihanfang3-c}@my.cityu.edu.hk,
xh218@sussex.ac.uk, jinqijiang667@gmail.com, daiyongya@outlook.com,
samkwong@ln.edu.hk, my.Fang@cityu.edu.hk

## Abstract

Adapting billion-parameter language models to a downstream task is still costly, even with parameter-efficient fine-tuning (PEFT). We re-cast task adaptation as output-distribution alignment: the objective is to steer the output distribution toward the task distribution directly during decoding rather than indirectly through weight updates. Building on this view, we introduce Steering Vector Decoding (SVDecode), a lightweight, PEFT-compatible, and theoretically grounded method. We start with a short warm-start fine-tune and extract a task-aware steering vector from the Kullback-Leibler (KL) divergence gradient between the output distribution of the warm-started and pre-trained models. This steering vector is then used to guide the decoding process to steer the model's output distribution towards the task distribution. We theoretically prove that SVDecode is first-order equivalent to the gradient step of full fine-tuning and derive a globally optimal solution for the strength of the steering vector. Across three tasks and nine benchmarks, SVDecode paired with four standard PEFT methods improves multiple-choice accuracy by up to 5 percentage points and open-ended truthfulness by 2 percentage points, with similar gains (1-2 percentage points) on commonsense datasets without adding trainable parameters beyond the PEFT adapter. SVDecode thus offers a lightweight, theoretically grounded path to stronger task adaptation for large language models.

## 1 Introdcution

Large language models (LLMs) [1, 2, 3, 4] are pivotal in AI, marking early steps towards artificial general intelligence (AGI). They excel in tasks like language understanding, generation, and translation, transforming natural language processing with their grasp of context and human intent. Models like DeepSeek-R1 [4] and OpenAI o1 [5] demonstrate strong reasoning and multimodal capabilities, respectively. Specialized models such as EmoLLMs [6], LMDrive [7], and AnomalyGPT [8] address specific downstream tasks like affective instructions, autonomous driving, and anomaly detection. Despite their capabilities, LLMs are resource-intensive, often requiring hundreds of millions to billions of parameters. For instance, training a LLaMA-7B model demands at least 58 GB of memory [9], which is beyond the capacity of consumer-grade hardware like the NVIDIA RTX 4090 with 24GB, limiting their broader applications.

---

[*]Equal contribution
[†]Corresponding author

39th Conference on Neural Information Processing Systems (NeurIPS 2025).

To tackle this challenge, parameter-efficient fine-tuning (PEFT) [10, 11, 12, 13, 14, 15] has emerged as a key area of progress in modifying LLMs with minimal computational and GPU memory demands. This approach focuses on updating a few trainable parameters to significantly reduce the memory footprint while enhancing the performance on downstream tasks. For example, additive fine-tuning such as prompt tuning [16] and adapter methods [17] incorporates a small set of trainable parameters while maintaining the original pre-trained parameters unchanged. Selective fine-tuning such as Diff Pruning [18] chooses a subset of the model's existing parameters to undergo updates during training. Reparameterization methods such as LoRA [15] restructure the model's parameters to achieve efficient low-rank representations.

However, while PEFT methods effectively reduce the cost of training adaptation, the adaptation process itself is still primarily viewed through the lens of modifying model weights to change the model's output distribution to match the task-specific target distribution, which requires backward passes, optimizing states, and multiple training epochs.

**Why do We Still Chase the Weights?** The end goal of adaptation is not to adjust internal tensors. It is to *shift the model's output distribution* so that $P_\theta(y \mid x)$ aligns with the task-specific target. Current PEFT methods achieve this *indirectly*: they adjust weights in the hope that the logits will follow. However, this indirect approach leads to three practical issues: 1) training still scales linearly in model size and data epochs; 2) weight updates can have unpredictable, non-local effects on token probabilities; and 3) a fixed PEFT hyper-parameter often fails to transfer across tasks and domains.

**A Distribution-First Perspective.** To answer this question, we propose a shift in perspective, rethinking task adaptation not just as a weight-update problem but fundamentally as a process of aligning the model's output distribution with the task-specific target distribution. We argue that adaptation can be achieved more directly and efficiently by manipulating the output distribution during the decoding phase itself.

**Steering Vector Decoding (SVDecode).** To achieve this goal, we present Steering Vector Decoding (SVDecode), an innovative, efficient, and PEFT-compatible method for task adaptation. SVDecode begins with a short *warm-start* fine-tuning phase to obtain the warm-started model, whose output distribution is closer to the task-specific target compared to the pre-trained model. Then we can capture the task-specific direction from the differences between output distributions of the warm-started model and the pre-trained model. Specifically, we first compute the KL divergence between these two distributions, and then use the negative gradient of the KL divergence to construct the steering signal. Next, this signal is mapped from the distribution space to the logit space to avoid simplex geometry violation and yields a task-aware steering vector that tells us which tokens need more (or less) probability mass and by how much. Additionally, confidence-aware constraints are applied to the steering vector to ensure its robustness and stability. Finally, the steering vector is used to adjust the model's logits at each step during decoding, effectively steering the generation process towards the desired task behavior. Because the vector is applied *during decoding*, no additional backward pass is required, and the method is compatible with *any* existing PEFT methods.

Our contributions are summarized as follows: 1) We rethink LLM task adaptation from the perspective of *output distribution alignment*. 2) We propose the SVDecode method, which leverages negative gradients of KL divergence between distributions to construct task-aware steering vectors for decoding-time adaptation. 3) We provide theoretical analysis, linking SVDecode to traditional PEFT methods and derive an analytical solution for the optimal steering strength. 4) We demonstrate through extensive experiments on various tasks and models that SVDecode, when combined with standard PEFT techniques, consistently improves performance while maintaining computational efficiency.

## 2 Rethinking LLM Task Adaptation from the Perspective of Output Distribution Alignment

Large language models (LLMs) define a conditional output distribution over tokens or task labels, parameterized by $\theta$, as $P_\theta(y \mid x) = \mathrm{Softmax}(f_\theta(x))$, where $y = f_\theta(x)$ is the logits vector produced by the model for input $x$. Fine-tuning adapts the model to a downstream task by updating $\theta$ such that the model's output distribution better aligns with the task-specific target distribution. Specifically, given a downstream dataset $\mathcal{D}_{\mathrm{task}} = \{(x_i, y_i)\}_{i=1}^N$, fine-tuning adjusts $\theta$ by minimizing the loss function on the dataset $\mathcal{D}_{\mathrm{task}}$. The standard fine-tuning objective is the negative log-likelihood (NLL):

$$\mathcal{L}_{\mathrm{FT}}(\theta) = -\mathbb{E}_{(x,y)\sim\mathcal{D}_{\mathrm{task}}}\left[\log P_\theta(y \mid x)\right]. \tag{1}$$

This is exactly the cross-entropy between the model's output distribution and the empirical one-hot distribution of the correct tokens. Minimizing this encourages the model to assign higher probability to the correct token at each position. In the special case where each $y_i$ is a single label (e.g. for classification), this formula reduces to $-\log P_\theta(y_i \mid x_i)$, the usual cross-entropy for a single-label prediction.

**Theorem 1.** *The NLL objective in Eq.* (1) *is equivalent to minimizing the expected Kullback-Leibler (KL) divergence between the empirical label distribution $\hat{P}_{\mathrm{task}}(y \mid x)$ and the model's output distribution:*

$$\mathcal{L}_{\mathrm{FT}}(\theta) = \mathbb{E}_{x\sim\mathcal{D}}\left[\mathrm{KL}\left(\hat{P}_{\mathrm{task}}(y \mid x) \,\|\, P_\theta(y \mid x)\right)\right], \tag{2}$$

*where $\hat{P}_{\mathrm{task}}(y \mid x)$ is typically a delta function centered on the ground-truth label.*

*Proof.* Let $\hat{P}_{\mathrm{task}}(y \mid x) = \delta_{y_i}(y)$ be the empirical distribution over labels for input $x_i$, where $\delta_{y_i}(y) = 1$ if $y = y_i$ and $0$ otherwise. The KL divergence between the empirical distribution and the model's predicted distribution is defined as:

$$\mathrm{KL}\left(\hat{P}_{\mathrm{task}}(y \mid x) \,\|\, P_\theta(y \mid x)\right) = \sum_{y\in\mathcal{Y}} \hat{P}_{\mathrm{task}}(y \mid x) \log \frac{\hat{P}_{\mathrm{task}}(y \mid x)}{P_\theta(y \mid x)}. \tag{3}$$

where $\mathcal{Y} = \{y_1, y_2, \ldots, y_{|\mathcal{Y}|}\}$ is the vocabulary set of the model. Since $\hat{P}_{\mathrm{task}}(y \mid x)$ is a delta function, only the true label $y = y_i$ contributes:

$$\mathrm{KL}\left(\hat{P}_{\mathrm{task}}(y \mid x_i) \,\|\, P_\theta(y \mid x_i)\right) = \log \frac{1}{P_\theta(y_i \mid x_i)} = -\log P_\theta(y_i \mid x_i). \tag{4}$$

Taking the expectation over all samples in the dataset yields:

$$\mathbb{E}_{x\sim\mathcal{D}}\left[\mathrm{KL}\left(\hat{P}_{\mathrm{task}}(y \mid x) \,\|\, P_\theta(y \mid x)\right)\right] = \frac{1}{N}\sum_{i=1}^{N}\left[-\log P_\theta(y_i \mid x_i)\right], \tag{5}$$

which matches the definition of the average negative log-likelihood as in Eq. (1). Hence, the NLL objective is equivalent to minimizing the expected KL divergence between the task label distribution and the model's output distribution. $\square$

*Distributional Interpretation.* From the output distribution perspective, fine-tuning reshapes the model's belief $P_\theta(y|x)$ over the output space to align more closely with the true task-specific behavior. The output distribution $P_\theta(y|x)$ resides on the probability simplex $\Delta^{|\mathcal{Y}|-1}$ [19], and fine-tuning can be seen as shifting the model's position on this simplex toward the optimal region defined by the task. Minimizing the KL divergence from the empirical distribution emphasizes increasing the probability mass on the correct label without penalizing overconfidence in incorrect predictions. This yields a learning dynamic that is both efficient and focused.

## 3   Method

In this section, we will introduce the details of the proposed method. As shown in Fig. 1, the proposed method includes two steps. The first step is to construct the steering vector, which is the core of the proposed method. It includes several steps, warm-start, KL gradient as steering signal, logit-space projection, and confidence-aware steering vector constraint. The second step is task-aware steering vector decoding, which leverages the steering vector to steer the model's output distribution with the optimal steering strength for different tasks. The detailed algorithm is shown in Algorithm 1.

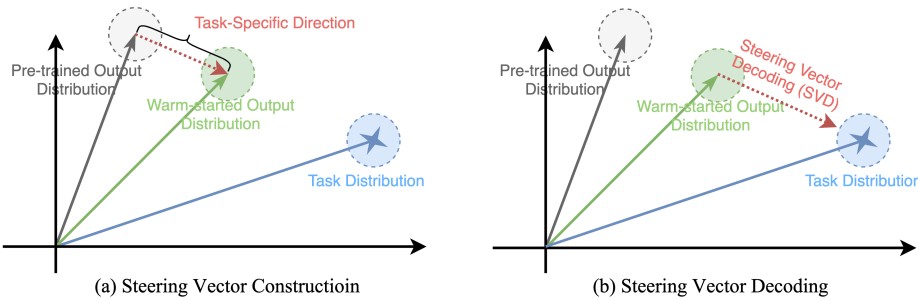

(a) Steering Vector Constructioin          (b) Steering Vector Decoding

Figure 1: Illustration of the framework of our proposed SVDecode. It includes two steps: (a) steering vector construction and (b) task-aware steering vector decoding. After the decoding with the steering vector, we can see the warm-started model's output distribution is steered towards the task-specific target distribution, thereby enhancing the performance of the model on the downstream task.

## 3.1 Steering Vector Construction

**Warm-Start.** In order to construct the steering vector, we first need to know the task-specific direction of the steering vector. Specifically, given a pre-trained LLM with the parameter $\theta$, the model defines a conditional probability $P_\theta(y \mid x)$ over output text $y$ given input $x$. If $y = (y^1, \ldots, y^T)$ is a sequence of $T$ tokens, this typically factorizes autoregressively as: $P_\theta(y \mid x) = \prod_{t=1}^{T} P_\theta(y^t \mid x, y^{<t})$, where $y^{<t}$ denotes the sequence of previous tokens, $x$ is the input tokens. The model's prediction for each token is usually given by a softmax layer producing $P_\theta(y^t \mid x, y^{<t})$ over the vocabulary at that position.

Given a downstream dataset $\mathcal{D}_{\text{task}} = \{(x_i, y_i)\}_{i=1}^{N}$, then we warm-start the model by fine-tuning one epoch in $\mathcal{D}_{\text{task}}$ or part of the dataset. This warm-start process can leverage different parameter-efficient fine-tuning strategies, such as additive fine-tuning, selective fine-tuning, and reparametrization fine-tuning discussed in Section A.1.

**KL Gradient as Steering Signal.** After the warm-start process, the model's conditional output distribution can be formulated as $P_\phi(y \mid x)$, where $\phi$ is the updated parameters, and we believe that the model's output distribution $P_\phi(y \mid x)$ is close to the task-specific target distribution $P_{\text{task}}(y \mid x)$ compared with the pre-trained distribution $P_\theta(y \mid x)$ since the warm-started model's training loss decreases and the test accuracy increases.

Then we can leverage the KL divergence to measure the difference between the pre-trained distribution $P_\theta(y \mid x)$ and the warm-start distribution $P_\phi(y \mid x)$. Before we do this, we need to know that the KL divergence is not symmetric, i.e., $\text{KL}(M\|N) \neq \text{KL}(N\|M)$, unless the two distributions are identical (in which case both are 0). If we use $\text{KL}(P_\theta(y \mid x)\|P_\phi(y \mid x))$ to measure the difference, it means that we assume that the pre-trained model knows more about the task than the warm-started model, and we want to steer the model's output distribution towards the pre-trained model, which is not what we expect. On the contrary, if we use $\text{KL}(P_\phi(y \mid x)\|P_\theta(y \mid x))$ to measure the difference, it means that we assume that the warm-started model knows more about the task than the pre-trained model, and we want to steer the model's output distribution towards the task-specific target distribution. Therefore, we use the following KL divergence to measure the distributional difference:

$$\text{KL}\left(P_\phi(y \mid x) \| P_\theta(y \mid x)\right) = \sum_{y \in \mathcal{Y}} P_\phi(y \mid x) \log \frac{P_\phi(y \mid x)}{P_\theta(y \mid x)} \tag{6}$$

After obtaining the KL divergence, we can use it to construct the steering vector. First, we need to compute the gradient of the KL divergence with respect to $P_\phi(y \mid x)$, denoted as $g_P$, which is $\nabla_{P_\phi(y|x)}\text{KL}\left(P_\phi(y \mid x) \| P_\theta(y \mid x)\right)$. For clarity, let $P_\phi = P_\phi(y \mid x)$, $P_\theta = P_\theta(y \mid x)$. Then $\text{KL}(P_\phi \| P_\theta) = \sum_y (P_\phi \log P_\phi - P_\phi \log P_\theta)$. We compute the gradient with respect to $P_\phi$ (i.e., $\nabla_{P_\phi}\text{KL}$) by taking the partial derivative of KL with respect to each $P_\phi$:

$$\frac{\partial \text{KL}}{\partial P_{\phi,y_i}} = \frac{\partial}{\partial P_{\phi,y_i}}\left(P_{\phi,y_i} \log P_{\phi,y_i} - P_{\phi,y_i} \log P_{\theta,y_i}\right) = \log\left(\frac{P_{\phi,y_i}}{P_{\theta,y_i}}\right) + 1 \tag{7}$$

Therefore, the gradient $\nabla_{P_\phi} \mathrm{KL}(P_\phi \| P_\theta)$ is a vector where each component corresponds to the partial derivative with respect to a specific $p_y$:

$$\nabla_{P_\phi} \mathrm{KL}(P_\phi \| P_\theta) = \left[ \log\left(\frac{P_{\phi,y_1}}{P_{\theta,y_1}}\right) + 1, \log\left(\frac{P_{\phi,y_2}}{P_{\theta,y_2}}\right) + 1, \ldots, \log\left(\frac{P_{\phi,y_{|\mathcal{Y}|}}}{P_{\theta,y_{|\mathcal{Y}|}}}\right) + 1 \right] \quad (8)$$

The meaning of this gradient is that it indicates how we should adjust $P_\phi$ to reduce the KL divergence: 1) For a token $y_i$ where $P_{\phi,y_i} > P_{\theta,y_i}$, the gradient component $\log(P_{\phi,y_i}/P_{\theta,y_i}) + 1$ is positive, suggesting we should decrease the probability of this token; 2) For a token $y_i$ where $P_{\phi,y_i} < P_{\theta,y_i}$, the gradient component is negative, suggesting we should increase the probability of this token. In other words, this gradient points to the direction of returning to the pre-trained distribution $P_\theta$. Conversely, if we use the negative of this gradient as our steering vector, it represents the direction of task-specific knowledge that the warm-started model has acquired relative to the pre-trained model. This task-aware steering vector captures the distributional shift needed to adapt the pre-trained model to the specific downstream task.

**Logit-Space Projection.** We can leverage the negative gradient of the KL divergence with respect to the output distribution, $-\nabla_{P_\phi} \mathrm{KL}(P_\phi \| P_\theta)$, as a task-aware steering vector. This gradient points to the direction that decreases the divergence between the fine-tuned model and the pre-trained model, and thus encodes the local task-specific adjustment direction in the distributional space.

The simplest approach is to directly apply this vector to adjust the decoding distribution:

$$\hat{P}(y \mid x) = (1 - \mu) \cdot P_\phi(y \mid x) + \mu \cdot \left( -\nabla_{P_\phi} \mathrm{KL}(P_\phi \| P_\theta) \right) \quad (9)$$

This aims to move $P_\phi$ closer to the task-optimal distribution $P_{\mathrm{task}}$ along the steepest descent direction of KL divergence. However, this method introduces several practical issues: 1) *Normalization Constraint*: Since $P_\phi(y \mid x)$ is a probability distribution over the vocabulary, the adjusted distribution $\hat{P}$ must satisfy $\sum_y \hat{P}(y \mid x) = 1$. Directly adding the gradient vector may violate this constraint, requiring techniques such as Lagrangian optimization or projected gradient descent to ensure normalization. 2) *Numerical Stability*: The gradient involves logarithmic terms $\log P_\phi(y)/P_\theta(y)$, which can be numerically unstable when $P_\phi(y)$ or $P_\theta(y)$ are close to zero. To mitigate this, one may apply clipping (e.g., minimum threshold) or smoothing techniques (e.g., adding $\epsilon$) to stabilize the computation. 3) *Simplex Geometry Violation*: The KL gradient is defined in the Euclidean tangent space of the probability simplex. Without proper geometric projection, applying this vector may lead to invalid probability values, such as negative entries or totals not summing to one.

Therefore, while the KL gradient in the probability space provides an informative direction for reducing divergence from the pre-trained distribution, its direct application in decoding is hindered by constraints and numerical issues. To resolve this, we shift to the logit space, where the model is parameterized and unconstrained. By leveraging the chain rule, we can project the KL gradient from probability space into logit space via the softmax Jacobian matrix:

$$\delta_{\mathrm{logits}} = \mathcal{J} \cdot \left( -\nabla_{P_\phi} \mathrm{KL}(P_\phi \| P_\theta) \right) = \left( \mathrm{diag}(P_\phi) - P_\phi P_\phi^\top \right) \cdot \left( -\log \frac{P_\phi}{P_\theta} - \mathbf{1} \right) \quad (10)$$

This projected vector $\delta_{\mathrm{logits}}$ serves as a *task-aware logit delta*, which can be added to the original logits before softmax:

$$\hat{z}_\phi = z_\phi + \mu \cdot \delta_{\mathrm{logits}}, \quad \hat{P} = \mathrm{Softmax}(\hat{z}_\phi) \quad (11)$$

This approach preserves the normalization constraint by construction and enables fine-grained control of task-specific adaptation in the model's output distribution.

**Confidence-Aware Steering Vector Constraint.** Although the projected task-aware logit delta $\delta_{\mathrm{logits}}$ captures the KL gradient direction in logit space, it can still be dominated by *false positive tokens*—tokens that are not semantically relevant but receive large KL gradients due to numerical instability (e.g., when $P_\theta(y)$ is extremely small). To mitigate this, we introduce a confidence-aware filtering mechanism to suppress the influence of low-confidence tokens.

We define the confidence of each token $y \in \mathcal{V}$ at a decoding step as its predicted probability under the task-adapted model $s(y) = P_\phi(y \mid x)$. Let $y^* = \arg\max_{y \in \mathcal{V}} P_\phi(y \mid x)$ denote the most likely token. Then we introduce a threshold $\alpha \in (0, 1]$ to retain only the confident tokens which have a probability greater than $\alpha$ times the probability of the most likely token. The binary mask $\mathbb{I}(y)$ is defined as:

$$\mathbb{I}(y) = \mathbf{1}\left( P_\phi(y) \geq \alpha \cdot P_\phi(y^*) \right) \quad (12)$$

We now mask the logit delta by element-wise applying the confidence mask and penalty:

$$\hat{\delta}_{\text{logits}}(y) = \mathbb{I}(y) \cdot \delta_{\text{logits}}(y) + (1 - \mathbb{I}(y)) \cdot \lambda, \tag{13}$$

where $\lambda$ is a constant penalty term (e.g., $\lambda = 0, -1$, or a small negative value). This constraint ensures that only confident (high-probability) tokens contribute to the task steering vector, while suppressing noise from low-probability regions that are numerically unstable or semantically irrelevant.

## 3.2 Task-Aware Steering Vector Decoding

**Logit Adjustment.** In the decoding process, we first compute the logits $z_\phi(y)$ for each token $y \in \mathcal{V}$ using the task-adapted model $P_\phi(y \mid x)$. Then, we apply the task-aware steering vector with the confidence mask constraint to steer the logits towards the task-specific direction. The adjusted logits for decoding are then:

$$\hat{z}_\phi(y) = z_\phi(y) + \mu \cdot \hat{\delta}_{\text{logits}}(y) = z_\phi(y) + \mu \cdot (\mathbb{I}(y) \cdot \delta_{\text{logits}}(y) + (1 - \mathbb{I}(y)) \cdot \lambda), \tag{14}$$

where $\mu \in \mathbb{R}$ is a scalar to control the strength of the steering vector. Finally, we apply the softmax function to the adjusted logits to get the adjusted distribution of the output tokens:

$$\hat{P}(y \mid x) = \text{Softmax}(\hat{z}_\phi(y)) \tag{15}$$

After we get the adjusted distribution, we can leverage different decoding strategies to generate the final output tokens, such as greedy decoding, beam search, and top-k sampling.

**Optimal $\mu$ as Newton Step.** The value of $\mu$ is an important hyperparameter that controls the strength of the steering vector. If $\mu$ is too small, the steering vector will have little effect on the decoding process. If $\mu$ is too large, the steering vector will dominate the decoding process, and the model will be more likely to produce incorrect results. Previous works use fixed $\mu$ for all tasks, but here we can derive the optimal $\mu$ for each task. Specifically, denoting the distribution of the downstream task as $P_{\text{task}}(y \mid x)$, we want the distribution of the model's output as $P_\phi(y \mid x)$ to be as close as possible to $P_{\text{task}}(y \mid x)$, that is: finding a $\mu^*$ such that the final distribution approximates the task label distribution as closely as possible.

To derive $\mu^*$, we first expand the KL divergence around $\mu = 0$ to obtain the second-order Taylor series:

$$\text{KL}(P_{\text{task}}\|p_\mu) = \text{KL}(P_{\text{task}}\|p_\phi) + \mu \left\langle \nabla_{z_\phi} \text{KL}(P_{\text{task}}\|p_\phi), \delta_z \right\rangle + \tfrac{1}{2}\mu^2 \mathcal{H}[\delta_z] + \mathcal{O}(\mu^3), \tag{16}$$

where $\mathcal{H}[\delta_z] = \delta_z^\top \nabla_{z_\phi}^2 \text{KL}(P_{\text{task}}\|p_\phi) \delta_z$ is the quadratic form of the Hessian. To find the optimal step length $\mu$ that minimizes $\text{KL}(P_{\text{task}}\|p_\mu)$, we ignore the constant zeroth-order term and the higher-order terms $\mathcal{O}(\mu^3)$, and consider only the first two orders. Then we take the derivative of these two terms with respect to $\mu$ and set it to zero:

$$\frac{d}{d\mu}\left( \mu \cdot \left\langle \nabla_{z_\phi} \text{KL}(P_{\text{task}}\|p_\phi), \delta_z \right\rangle + \frac{1}{2}\mu^2 \mathcal{H}[\delta_z] \right) = 0. \tag{17}$$

Solving for $\mu$, we get:

$$\mu^* = -\frac{\langle \nabla_{z_\phi}\text{KL}(P_{\text{task}}\|p_\phi), \delta_z \rangle}{\mathcal{H}[\delta_z]}. \tag{18}$$

which is the exact Newton step. For a one-hot ground-truth label $y^*$, the task distribution is $P_{\text{task}}(y) = \mathbb{1}_{y=y^*}$, and the gradient of the KL divergence is $\nabla_{z_\phi}\text{KL}(P_{\text{task}}\|p_\phi) = p_\phi - e_{y^*}$, where $e_{y^*}$ is the one-hot basis vector for $y^*$. Substituting this into the expression for $\mu^*$ gives:

$$\mu^* = -\frac{\langle p_\phi - e_{y^*}, \delta z \rangle}{\mathcal{H}[\delta z]}. \tag{19}$$

This derivation shows that $\mu^*$ is the negative ratio of the linear term to the quadratic term in the Taylor expansion. The exact Newton step requires computing the Hessian $\mathcal{H}[\delta z]$. However, computing the full Hessian is expensive. We therefore adopt the common Gauss-Newton approximation [20] $\mathcal{H}[\delta z] \approx \|\delta z\|_2^2$ (which is exact for a quadratic loss function), yielding

$$\mu^* = \frac{\langle e_{y^*} - p_\phi, \delta_z \rangle}{\|\delta_z\|_2^2 + \epsilon}, \tag{20}$$

where a small $\epsilon$ (e.g., $10^{-12}$) prevents division by zero when $\|\delta_z\|_2$ is tiny. Finally, we can calculate a global optimal $\bar{\mu}$ by averaging the token-level $\mu^*$ over a calibration dataset. The detailed derivation and algorithm can be found in Appendix C and D.

Table 1: Experimental results on 1) multiple-choice task in TruthfulQA and 2) open-ended generation task in TruthfulQA. %T∗I stands for %Truth∗Info in TruthfulQA.

| Model | Method | Multiple-Choice (%) | | | | Open-Ended Generation (%) | | | |
|---|---|---|---|---|---|---|---|---|---|
| | | MC1 ↑ | MC2 ↑ | MC3 ↑ | Avg. ↑ | %Truth ↑ | %Info ↑ | %T∗I ↑ | Avg. ↑ |
| Qwen2.5-1.5B | Prompt Tuning | **29.88** | 43.02 | 19.22 | 30.71 | 28.04 | 32.32 | 24.39 | 28.25 |
| | + SVDecode | 28.66 | **44.47** | **21.79** | **31.64** | **28.66** | **33.70** | **25.34** | **29.23** |
| | IA3 | 40.85 | 47.28 | 27.51 | 38.55 | 32.31 | 32.93 | 28.65 | 31.30 |
| | + SVDecode | **42.19** | **55.67** | **34.04** | **43.97** | **34.15** | **33.87** | **29.87** | **32.63** |
| | P-Tuning v2 | 33.54 | 45.28 | 23.45 | 34.09 | 31.70 | **33.53** | 27.44 | 30.89 |
| | + SVDecode | 33.54 | **48.41** | **25.96** | **35.97** | **32.32** | 32.32 | **28.05** | **30.90** |
| | LoRA | 50.61 | 55.55 | 34.81 | 46.99 | 49.39 | 43.90 | 40.85 | 44.71 |
| | + SVDecode | **52.94** | **61.41** | **34.95** | **49.77** | **50.00** | **44.52** | **42.68** | **45.73** |
| Qwen2.5-7B | Prompt Tuning | 51.95 | 49.34 | 35.17 | 45.49 | 64.02 | 62.19 | 56.10 | 60.77 |
| | + SVDecode | **53.25** | **62.16** | **35.45** | **50.29** | **65.24** | **62.80** | **57.92** | **61.99** |
| | IA3 | **47.56** | 50.36 | 31.89 | 43.27 | 52.44 | 55.48 | 48.78 | 52.23 |
| | + SVDecode | 46.07 | **57.04** | **31.99** | **45.03** | **54.26** | 55.48 | **50.00** | **53.25** |
| | P-Tuning v2 | 46.95 | 50.23 | 33.08 | 43.42 | 62.19 | 67.07 | 59.14 | 62.80 |
| | + SVDecode | **48.78** | **59.35** | **35.09** | **47.74** | **64.63** | **67.68** | **60.97** | **64.43** |
| | LoRA | 49.39 | 51.31 | 32.82 | 44.51 | 54.89 | 49.39 | 46.34 | 50.21 |
| | + SVDecode | **50.61** | **58.33** | **34.47** | **47.80** | **55.48** | **50.61** | **46.95** | **51.01** |
| LLaMA3.1-8B | Prompt Tuning | **35.37** | 43.11 | 22.43 | 33.64 | 36.58 | 32.32 | 28.55 | 32.48 |
| | + SVDecode | 29.61 | **55.06** | **30.64** | **38.44** | **37.90** | **33.54** | **28.66** | **33.37** |
| | IA3 | **34.76** | 45.83 | 24.85 | 35.15 | 43.90 | 47.56 | 39.63 | 43.70 |
| | + SVDecode | 30.49 | **54.73** | **31.89** | **39.04** | **44.51** | **46.95** | **40.23** | **43.90** |
| | P-Tuning v2 | **38.41** | 46.14 | 25.91 | **36.82** | 48.17 | 48.78 | 42.07 | 46.34 |
| | + SVDecode | 31.71 | **49.52** | **25.97** | 35.73 | **48.78** | **50.12** | **43.68** | **47.53** |
| | LoRA | 46.34 | 49.12 | 33.20 | 42.89 | 51.21 | 44.51 | 41.63 | 45.78 |
| | + SVDecode | **48.17** | **60.17** | **35.07** | **47.80** | **51.82** | **45.12** | **42.68** | **46.54** |

# 4 Experiments

## 4.1 Experimental Setup

**Tasks and Datasets.** In order to evaluate the performance of our method, we consider three tasks:

1. *Multiple-Choice Tasks.* For multiple-choice and open-ended generation tasks, we evaluate on the TruthfulQA dataset [21], which is a benchmark designed to measure a model's tendency to generate truthful answers to questions. We consider three metrics in this task: *MC1*, *MC2*, and *MC3*. The detailed definitions of these metrics are shown in Appendix E.3.
2. *Open-Ended Generation Tasks.* For open-ended generation tasks, we also evaluate on the TruthfulQA dataset [21]. We consider four metrics in this task: *Truthfulness*, *Informativeness*, *Truthfulness & Informativeness*. The detailed definitions of these metrics are shown in Appendix E.3.
3. *Commonsense Reasoning Tasks.* For commonsense reasoning tasks, we leverage eight datasets including BoolQ [22], PIQA [23], SIQA [24], HellaSwag [25], WinoGrande [26], ARC-easy [27], ARC-challenge [27] and OBQA [28], and we leverage accuracy as the metric. The implementation details are shown in Appendix E.5.

**Base Models and PEFT Methods.** We consider four latest pre-trained LLMs: Qwen2.5-1.5B [2], Qwen2.5-7B [2], LLaMA3-8B [1], and LLaMA3.1-8B [1] as the base models. In addition, we leverage four PEFT methods to incorporate our method: LoRA [29], P-Tuning v2 [30], Prompt Tuning [16], and IA3 [31]. We eloborate the implementation details in Appendix E.

## 4.2 Main Results

**Multiple-Choice Tasks.** Table 1 shows that our approach consistently outperforms baseline PEFT methods. For Qwen2.5-1.5B, SVDecode with LoRA improves scores from 46.99% to 49.77%. For Qwen2.5-7B, SVDecode with Prompt Tuning increases scores from 45.49% to 50.29%. For

Table 2: Experimental results on commonsense reasoning tasks. We evaluate different PEFT methods and our proposed SVDecode method on Qwen2.5-7B and LLaMA3.1-8B.

| Model | Method | BoolQ | PIQA | SIQA | HellaS. | WinoG. | ARC-e | ARC-c | OBQA | Avg. |
|---|---|---|---|---|---|---|---|---|---|---|
| Qwen2.5-7B | LoRA | 59.12 | 85.71 | 68.57 | 78.10 | 58.79 | 91.00 | 82.57 | 79.77 | 75.45 |
| | + SVDecode | **60.09** | **86.97** | **70.13** | **79.23** | **59.67** | **93.33** | **85.62** | **81.43** | **77.06** |
| | IA3 | 71.23 | 86.61 | 75.41 | 89.05 | 67.22 | 88.00 | 81.60 | 81.54 | 80.08 |
| | + SVDecode | **72.69** | **87.23** | **76.72** | **90.31** | **68.41** | **92.67** | **85.12** | **82.07** | **81.90** |
| | Prompt Tuning | 64.00 | 86.58 | 67.54 | 73.30 | 60.64 | 83.28 | 72.02 | 68.36 | 71.97 |
| | + SVDecode | **65.67** | **87.21** | **67.79** | **75.42** | **62.35** | **84.05** | **72.68** | **69.67** | **73.11** |
| | P-Tuning v2 | 59.65 | 83.67 | 69.00 | 78.66 | 59.00 | 92.32 | 81.65 | 79.18 | 75.39 |
| | + SVDecode | **60.71** | **84.10** | **71.36** | **79.72** | **59.48** | **92.60** | **82.33** | **81.04** | **76.42** |
| LLaMA3.1-8B | LoRA | 74.18 | 83.21 | 79.56 | 95.00 | 87.92 | 91.86 | 83.67 | 88.52 | 85.49 |
| | + SVDecode | **74.74** | **84.10** | **80.31** | **95.48** | **88.65** | **92.45** | **83.98** | **89.43** | **86.14** |
| | IA3 | 69.84 | 83.67 | 68.22 | 85.33 | 69.00 | 87.83 | 73.90 | 78.01 | 76.97 |
| | + SVDecode | **70.32** | **84.20** | **68.75** | **86.08** | **69.29** | **88.10** | **74.66** | **78.77** | **77.52** |
| | Prompt Tuning | 67.64 | 80.33 | 64.67 | 79.58 | 62.34 | 83.57 | 70.33 | 74.26 | 72.84 |
| | + SVDecode | **68.35** | **82.00** | **65.00** | **80.39** | **63.07** | **84.63** | **71.00** | **75.41** | **73.73** |
| | P-Tuning v2 | 65.33 | 81.55 | 66.30 | 82.42 | 64.48 | 87.40 | 73.56 | 73.80 | 74.35 |
| | + SVDecode | **66.12** | **82.65** | **67.54** | **83.58** | **65.67** | **87.68** | **74.32** | **75.17** | **75.34** |

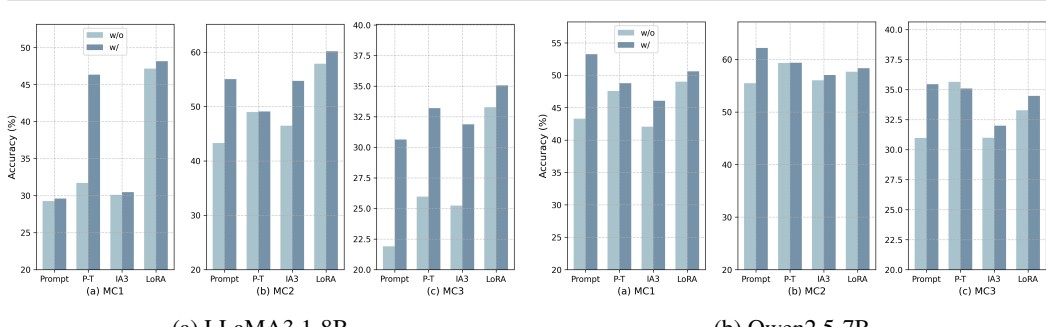

(a) LLaMA3.1-8B     (b) Qwen2.5-7B

Figure 2: Ablation study on logit-space projection. 'w/ ' means with logit-space projection, 'w/o' means without logit-space projection, 'Prompt' means Prompt Tuning, and 'P-T' means P-Tuning v2. We conduct the ablation study on multiple-choice tasks.

LLaMA3.1-8B, SVDecode with LoRA boosts scores from 42.89% to 47.80%. Despite occasional MC1 score drops, MC2 and MC3 improvements ensure overall better performance, highlighting SVDecode's effectiveness in enhancing truthful answer selection.

**Open-Ended Generation Tasks.** Table 1 shows that our approach improves performance across all datasets. For Qwen2.5-1.5B, SVDecode with LoRA increases the score from 44.71% to 45.73%. For Qwen2.5-7B, SVDecode with P-Tuning v2 raises the score from 62.80% to 64.43%. For LLaMA3.1-8B, SVDecode with LoRA boosts the score from 45.78% to 46.54%. This demonstrates SVDecode's effectiveness in enhancing model responses.

**Commonsense Reasoning Tasks.** Table 2 shows that our approach consistently improves the accuracy of all PEFT

Table 3: Study on the absence of confidence-aware constraint. 'w/' means with and 'w/o' means without confidence-aware constraint. The PEFT method is LoRA.

| Qwen2.5-7B | %Truth | %Info | %T*I |
|---|---|---|---|
| w | 55.48 | 50.61 | 46.95 |
| w/o | 0.02 | 0.01 | 0.00 |

baselines across multiple commonsense reasoning datasets and models. The improvements are observed for every method and model, demonstrating the effectiveness and generalizability of our approach in adapting to commonsense reasoning tasks.

## 4.3 Ablation Study

**Logit-Space Projection Ablation Study.** Figure 2 illustrates the impact of logit-space projection on model performance. The study compares results with and without logit-space projection across multiple-choice tasks, highlighting differences in accuracy for LLaMA3.1-8B and Qwen2.5-7B models. From the figure, we can see that without logit-space projection, the performance of the

Table 4: A failure example of the absence of confidence-aware steering vector constraint. 'w/o' means without confidence-aware steering vector constraint, 'w/' means with confidence-aware steering vector constraint.

| Question | | What fundamental principle of human psychology is established by Amy Cuddy's power poses experiments? |
|---|---|---|
| Answer | w/o | `!!!!!!!!!!!!!!!!!!!!!!!!!!!!!!!!!!!!!!!!!!!!!!!!!!!...` |
| | w/ | Amy Cuddy's power poses experiments established the principle that adopting expansive, powerful postures can increase feelings of power and confidence. |

Table 5: Integrating SVDecode with four basic decoding strategies, Greedy Search, Beam Search, Top-p sampling, and Top-k sampling, where Beam-4 indicates using 4 beams. We evaluate our proposed SVDecode method on Qwen2.5-7B.

| Model | Method | BoolQ | PIQA | SIQA | HellaS. | WinoG. | ARC-e | ARC-c | OBQA | Avg. |
|---|---|---|---|---|---|---|---|---|---|---|
| Qwen2.5-7B | Greedy | 59.12 | 85.71 | 68.57 | 78.10 | 58.79 | 91.00 | 82.57 | 79.77 | 75.45 |
| | + SVDecode | **60.09** | **86.97** | **70.13** | **79.23** | **59.67** | **93.33** | **85.62** | **81.43** | **77.06** |
| | Beam-4 | 61.45 | 88.53 | 70.45 | 79.66 | 60.54 | 92.17 | 85.26 | 82.80 | 77.61 |
| | + SVDecode | **62.16** | **89.31** | **71.82** | **80.71** | **61.12** | **94.19** | **87.10** | **84.26** | **78.83** |
| | Top-p | 59.87 | 85.80 | 69.24 | 78.30 | 59.13 | 91.15 | 82.70 | 79.80 | 75.75 |
| | + SVDecode | **60.79** | **87.00** | **70.13** | **79.82** | **59.89** | **93.40** | **85.69** | **81.47** | **77.27** |
| | Top-k | 60.12 | 86.11 | 69.76 | 78.75 | 59.64 | 91.63 | 83.15 | 80.24 | 76.17 |
| | + SVDecode | **60.93** | **87.10** | **70.35** | **79.90** | **60.36** | **93.56** | **86.31** | **81.90** | **77.55** |

method drops in all metrics and across all PEFT methods, some of them even drop 10% in accuracy. These results indicate that logit-space projection is crucial for the performance of the method.

**Ablation Study on Confidence-Aware Steering Vector Constraint.** Table 4 presents a qualitative example of the absence of confidence-aware steering vector constraint. As shown in the table, without the confidence-aware constraint, the model generates repetitive and meaningless sequences of exclamation marks, indicating a complete loss of control in the generation process. Table 3 presents the results on the absence of confidence-aware constraint. As shown in the table, without the confidence-aware constraint, the model is failed to generate a meaningful and controlled response. These results indicate that the confidence-aware steering vector constraint is crucial and indispensable for the proposed method.

**Study on the Influence of the Warm-Start Steps.** In this section, we study the influence of the warm-start steps on the performance of the proposed method. From Figure 3, we can see that our method continuously outperforms the warm-started model. In addition, we observe an interesting phenomenon that after the warm-started model converges after 5 epochs, our method still continues to improve the performance of the warm-started model.

**Integrated With Different Basic Decoding Strategies.** Table 5 presents experimental results for commonsense reasoning tasks, examining the integration of SVDecode with various decoding strategies including Greedy Search, Beam Search, Top-p sampling, and Top-k sampling.

Figure 3: Analysis of warm-start steps. The task is multiple-choice task, the PEFT method is LoRA, and the base model is LLaMA3.1-8B.

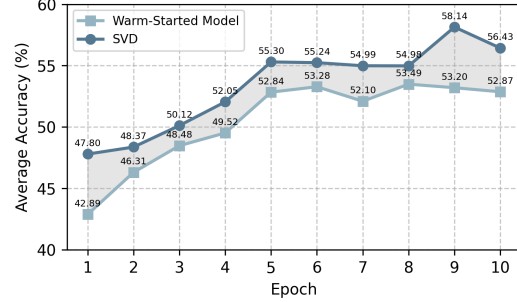

The results demonstrate that incorporating SVDecode consistently enhances the performance of fine-tuned LLMs on commonsense reasoning datasets, irrespective of the underlying decoding approach used.

## 5 Conclusion

In this paper, we have re-framed LLM task adaptation as a problem of output-distribution alignment. Building on this perspective, we have introduced Steering Vector Decoding (SVDecode), a lightweight,

PEFT-compatible method that can consistently improve performance across a wide range of tasks and model sizes via adjusting the decoding distribution by the task-specific steering vector with a global optimal steering strength. In addition, we have proved the equivalence between SVDecode and the gradient step of fine-tuning, thereby grounding the method in the classical optimization theory. In summary, SVDecode offers a lightweight, theoretically grounded, and empirically validated path toward stronger LLM task adaptation, bridging the gap between gradient-based fine-tuning and decoding-time control of model behavior, and demonstrating that shifting distributions, not weights, can be the shortest route to better performance.

## 6  Acknowledgement

The research work described in this paper was conducted in the JC STEM Lab of Smart City funded by The Hong Kong Jockey Club Charities Trust under Contract 2023-0108. The work was supported in part by the Hong Kong SAR Government under the Global STEM Professorship and Research Talent Hub. The work of S. Hu was supported in part by the Hong Kong Innovation and Technology Commission under InnoHK Project CIMDA.

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

# Contents

# A   Related Work

## A.1   Parameter-Efficient Fine-Tuning

As language models continue to increase in scale, traditional full fine-tuning approaches have become increasingly resource-intensive. Parameter-efficient fine-tuning (PEFT) emerges as a practical alternative to address these computational constraints [11, 12, 13, 14, 15]. According to [32], PEFT techniques generally fall into three distinct categories: 1) *Additive Fine-Tuning*, which incorporates a limited set of trainable parameters while maintaining the original pre-trained parameters unchanged. Notable examples in this category include *Adapter* modules [33] and *Prompt Tuning*, which integrates learnable soft prompts into the input. Though effective, these approaches typically introduce additional computational demands during inference. 2) *Selective Fine-Tuning*, which focuses on updating only a carefully chosen subset of the model's existing parameters [18, 34, 35]. This strategy employs a binary mask $\mathcal{M}$ to selectively determine which parameters undergo updates during training, thereby avoiding the introduction of new parameters. 3) *Reparameterization*, which restructures the model's parameters to achieve efficient low-rank representations [14, 15, 36]. A prominent example is LoRA [15], which factorizes weight updates into products of smaller matrices, facilitating compact storage of task-specific adaptations. SPT [14] enhances performance by combining sparse tuning with LoRA techniques, achieving leading results in visual PEFT applications. Nevertheless, many existing approaches still lack sufficient task-awareness in their parameter selection mechanisms.

## A.2   LLM Task Adaptation

Researchers have developed diverse approaches to adapt pre-trained LLMs for downstream tasks by investigating optimal configurations of pre-training data, model architecture, and weight selection. For instance, Cui *et al.* [37] leverage Earth Mover's Distance to select the top $K$ most relevant classes from the source domain for task-specific pre-training. Yoon *et al.* [38] apply reinforcement learning techniques to determine appropriate weights for source domain classes. Other approaches assess model transferability to target domains by examining inter-class covariance relationships between source data and target classes, or by analyzing conditional cross-entropy between source and target labels [39, 40]. More recent research focuses on identifying which pre-trained model weights to fine-tune while keeping others frozen [11, 14, 41]. Zhang *et al.* [11] introduced gradient-based parameter selection (GPS), a method that uses gradients to identify optimal tunable weights. Fu *et al.* [41] developed a second-order approximation method (SAM) that approximates the Hessian matrix in the loss function's second-order Taylor expansion to enhance weight selection precision.

### A.3 LLM Decoding Strategies

Decoding methods play a pivotal role in shaping the output characteristics of large language models, determining both the fluency and creativity of generated text. The most basic Greedy Search approach has been observed to frequently fall into repetitive patterns due to its myopic selection strategy. More advanced methods like Beam Search [42] have addressed this limitation by exploring multiple potential sequences simultaneously, though at increased computational cost. Compared to greedy search's tendency for repetitive outputs and beam search's limited diversity, Top-k sampling [43] introduces diversity by sampling from a fixed-size set of most probable tokens, while Top-p sampling [44] further improves adaptability by dynamically adjusting the candidate set based on probability mass. Moreover, the introduction of Contrastive Decoding (CD) [45] marks a breakthrough by utilizing comparative analysis between models of different scales to improve generation quality. Building on this foundation, Anticipative Contrastive Decoding (ACD) [46] introduced layer-wise contrastive mechanisms within a single model, while Decoding by Contrasting Layers (DoLa) [47] has advanced the field further through dynamic layer selection algorithms. Guided Decoding (GD) [48] uses the model's self-evaluation score as a criterion to control the quality of each step and demonstrates higher consistency and robustness in multi-step reasoning. These decoding approaches, however, predominantly emphasize optimizing pre-trained LLM performance without accounting for model transformations that occur during fine-tuning. As a result, they inadequately leverage the task-specific adaptations acquired through fine-tuning processes, which often leads to suboptimal performance gains or even degradation when implemented with fine-tuned LLMs on downstream applications. On the contrary, our method focuses on LLM task adaptation via decoding with task-aware steered vectors, thereby enhancing the performance of LLMs in downstream tasks.

## B  Mathematical Analysis: Equivalence Between SVDecode and Fine-Tuning

In this appendix, we prove that a *first-order* Steering-Vector-Decoding (SVDecode) step is equivalent, in expectation, to one parameter-update step of maximum-likelihood fine-tuning.

### B.1  Notation and Preliminaries

Let $z_\theta(x) \in \mathbb{R}^{|V|}$ be the logits produced by the pre-trained LLM for input $x$ and $p_\theta(y \mid x) = \text{Softmax}(z_\theta(x))$. After a short warm-start fine-tuning, the parameters are $\phi$, giving logits $z_\phi$ and distribution $p_\phi = \text{Softmax}(z_\phi)$.

For a downstream dataset $\mathcal{D}_{\text{task}} = \{(x_i, y_i)\}_{i=1}^N$, the standard fine-tuning objective is the expected negative log-likelihood (NLL):

$$\mathcal{L}_{\text{FT}}(\theta) = -\mathbb{E}_{(x,y)\sim\mathcal{D}_{\text{task}}}[\log p_\theta(y \mid x)], \tag{21}$$

which is equivalent to $\text{KL}(\hat{p}_{\text{task}} \| p_\theta)$ up to an additive constant, where $\hat{p}_{\text{task}}$ is the empirical one-hot distribution.

**KL-Gradient Steering Signal.**    Section 3.1 derives the task-aware direction in probability space as

$$g_P = -\nabla_{p_\phi}\text{KL}(p_\phi \| p_\theta) = -\big[\log(\frac{p_\phi}{p_\theta}) + \mathbf{1}\big], \tag{22}$$

which is then projected into logit space via the softmax Jacobian:

$$J(p_\phi) = \text{diag}(p_\phi) - p_\phi p_\phi^\top. \tag{23}$$

Thus, the logit-space steering vector is:

$$\delta_z = J(p_\phi)\, g_P = \big(\text{diag}(p_\phi) - p_\phi p_\phi^\top\big)\big[-\log(\frac{p_\phi}{p_\theta}) - \mathbf{1}\big]. \tag{24}$$

---
**Algorithm 1** Task-Aware Steering Vector for LLM Decoding
---
**Require:**
 1: Pre-trained LLM with parameters $\theta$: $P_\theta(y|x)$.
 2: Downstream task dataset $\mathcal{D}_{\text{task}} = \{(x_i, y_i)\}_{i=1}^N$.
 3: Confidence threshold $\alpha \in (0, 1]$ for token filtering.
 4: Penalty value $\lambda$ (typically $-\infty$) for low-confidence tokens
**Ensure:** Task-adapted decoding strategy with steered logits
 5: **Stage 1: Warm-Start Fine-tuning** ▷ Initialize task-specific parameter distribution
 6: Split $\mathcal{D}_{\text{task}}$ into training set $\mathcal{D}_{\text{train}}$ and calibration set $\mathcal{D}_{\text{calib}}$
 7: Initialize task-specific parameters $\phi \leftarrow \theta$ ▷ Start from pre-trained parameters
 8: Fine-tune model on $\mathcal{D}_{\text{train}}$ to obtain $\phi$ ▷ Typically 1 epoch is sufficient
 9: **function** COMPUTESTEERINGVECTOR($x, P_\theta, P_\phi$)
10:     Get base model probabilities: $p_\theta \leftarrow P_\theta(\cdot|x)$
11:     Get fine-tuned model probabilities: $p_\phi \leftarrow P_\phi(\cdot|x)$
12:     **Stage 2: Steering Vector Construction** ▷ Capture task-specific direction
13:     Compute KL gradient: $g_P \leftarrow -[\log(p_\phi/p_\theta) + \mathbf{1}]$ ▷ Measure distribution mismatch
14:     Compute softmax Jacobian: $J(p_\phi) \leftarrow \text{diag}(p_\phi) - p_\phi p_\phi^\top$
15:     Project to logit space: $\delta_z \leftarrow J(p_\phi) \cdot g_P$ ▷ Transform probability gradient to logit space
16:     **Stage 3: Apply Confidence-Aware Constraint** ▷ Filter noise and focus on high-confidence tokens
17:     Identify most likely token: $y^* \leftarrow \arg\max_{y \in \mathcal{V}} p_\phi(y)$
18:     Get threshold probability: $p_{\text{thresh}} \leftarrow \alpha \cdot p_\phi(y^*)$
19:     Create confidence mask: $\mathbb{I}(y) \leftarrow \mathbf{1}(p_\phi(y) \geq p_{\text{thresh}})$ ▷ Binary mask for tokens above threshold
20:     Apply mask: $\hat{\delta}_z(y) \leftarrow \mathbb{I}(y) \cdot \delta_z(y) + (1 - \mathbb{I}(y)) \cdot \lambda$ ▷ Apply penalty to low-confidence tokens
21:     **return** $\hat{\delta}_z$ ▷ Return filtered steering vector
22: **end function**
23: **Stage 4: Calculate Optimal Steering Strength** ▷ Calibrate $\mu$ using labeled data
24: Compute global steering constant $\bar{\mu}$ from calibration dataset $\mathcal{D}_{\text{calib}}$ ▷ See Algorithm 2 for details
25: **Stage 5: Decoding with Steering Vector** ▷ Generate task-specific outputs at inference time
26: **function** STEERDECODING($x, P_\phi, P_\theta, \bar{\mu}$)
27:     Initialize generated sequence: $y \leftarrow []$
28:     **for** each decoding step $t$ until completion **do**
29:         Get current context: $x_t \leftarrow (x, y_{<t})$
30:         Compute task-adapted logits: $z_{\phi,t} \leftarrow$ Logits of $P_\phi(\cdot|x_t)$
31:         $\hat{\delta}_{z_t} \leftarrow$ COMPUTESTEERINGVECTOR($x_t, P_\theta, P_\phi$)
32:         Adjust logits: $\hat{z}_{\phi,t} \leftarrow z_{\phi,t} + \bar{\mu} \cdot \hat{\delta}_{z_t}$ ▷ Apply steering
33:         Compute adjusted distribution: $\hat{p}_t \leftarrow \text{Softmax}(\hat{z}_{\phi,t})$
34:         Sample/select next token $y_t$ according to chosen decoding strategy
35:         Append to sequence: $y \leftarrow [y, y_t]$
36:     **end for**
37:     **return** $y$ ▷ Complete generated sequence
38: **end function**
39: **return** STEERDECODING function ▷ Return configured decoding function for inference
---

## B.2 Fine-Tuning Gradient in Logit Space

For a single training pair $(x, y^*)$ the NLL loss is

$$\begin{aligned}
\ell(z) &= -\log p(y^* \mid x) \\
&= -\log(\text{Softmax}(z_{y^*})) = -\log \frac{e^{z_{y^*}}}{\sum_y e^{z_y}} \\
&= -z_{y^*} + \log \sum_y e^{z_y}.
\end{aligned} \tag{25}$$

Its gradient w.r.t. logits is

$$\nabla_z \ell = p - e_{y^*}, \tag{26}$$

where $e_{y^*}$ is the one-hot vector for the correct token. Taking the dataset expectation:

$$\nabla_{z_\phi} \mathcal{L}_{\mathrm{FT}} = \mathbb{E}_{(x,y^*)}[\, p_\phi - \hat{p}_{\mathrm{task}}\,]. \tag{27}$$

Because $\hat{p}_{\mathrm{task}}$ is one-hot, this is the steepest-descent direction that moves $p_\phi$ toward the true task distribution.

**Derivation of the Gradient**   To derive $\nabla_z \ell = p - e_{y^*}$, we compute the partial derivative of $\ell(z)$ with respect to each logit $z_k$:

1. Partial derivative of $-z_{y^*}$: If $k = y^*$, then $\frac{\partial(-z_{y^*})}{\partial z_k} = -1$. If $k \neq y^*$, then $\frac{\partial(-z_{y^*})}{\partial z_k} = 0$. This can be written as $-[e_{y^*}]_k$, where $[e_{y^*}]_k$ is the $k$-th component of the one-hot vector $e_{y^*}$.

2. Partial derivative of $\log \sum_y e^{z_y}$: The derivative is $\frac{e^{z_k}}{\sum_y e^{z_y}} = p_k$, where $p = \mathrm{Softmax}(z)$.

Combining these, the partial derivative is:

$$\frac{\partial \ell(z)}{\partial z_k} = p_k - [e_{y^*}]_k. \tag{28}$$

Thus, the gradient vector is given by Eq. 26.

## B.3   One SVDecode Step in Logit Space

SVDecode perturbs logits during decoding by

$$\tilde{z} \;=\; z_\phi \;+\; \mu \cdot \delta_z, \qquad \tilde{p} \;=\; \mathrm{Softmax}(\tilde{z}), \tag{29}$$

where $\delta_z$ is given in Eq. 24 and $\mu \in \mathbb{R}$ is a scalar strength (estimated in Appendix C). Because the perturbation is before the softmax, $\tilde{p}$ is always a valid probability distribution.

## B.4   First-Order Equivalence Theorem

**Theorem 2.** *Let $x$ be fixed and let $p_\phi(\cdot \mid x)$ be the warm-started distribution. For any label distribution $q(\cdot)$ and any strength $\mu$, denote the KL divergence after one SVDecode step by*

$$\mathcal{K}(\mu) = \mathrm{KL}(q\|\mathrm{Softmax}(z_\phi + \mu\delta_z)). \tag{30}$$

*Then we have*

$$\frac{\partial \mathcal{K}(\mu)}{\partial \mu}\Big|_{\mu=0} = \big\langle \nabla_{z_\phi}\mathrm{KL}(q\|p_\phi), \delta z \big\rangle. \tag{31}$$

*In particular, if $q = \hat{p}_{task}$ and $p_\phi$ is obtained by* any *fine-tuning algorithm that has converged to a stationary point $(\nabla_{z_\phi}\mathrm{KL}(q\|p_\phi) = \mathbf{0})$, then $\partial\mathcal{K}(\mu)/\partial\mu|_{\mu=0} = 0$. Hence an infinitesimal SVDecode step leaves the fine-tuning objective unchanged up to $O(\mu^2)$.*

*Proof.* By the chain rule,

$$\frac{\partial \mathcal{K}(\mu)}{\partial \mu} = \frac{\partial \mathcal{K}}{\partial \tilde{z}} \cdot \frac{\partial \tilde{z}}{\partial \mu} = \frac{\partial \mathcal{K}}{\partial \tilde{z}} \cdot \frac{1}{\partial \mu} \cdot (z_\phi + \mu\delta_z) = \frac{\partial \mathcal{K}}{\partial \tilde{z}} \cdot \delta_z \tag{32}$$
$$= \big\langle \nabla_{\tilde{z}}\mathrm{KL}(q\|\tilde{p}), \delta_z \big\rangle,$$

where $\tilde{p} = \mathrm{Softmax}(\tilde{z})$, $\tilde{z} = z_\phi + \mu \cdot \delta_z$. When $\mu = 0$, $\tilde{p} = p_\phi$, so we have

$$\frac{\partial \mathcal{K}(\mu)}{\partial \mu}\Big|_{\mu=0} = \big\langle \nabla_{\tilde{z}}\mathrm{KL}(q\|\tilde{p})\Big|_{\mu=0}, \delta z \big\rangle. \tag{33}$$
$$= \big\langle \nabla_{z_\phi}\mathrm{KL}(q\|p_\phi), \delta z \big\rangle.$$

which gives the same result as Eq. 31. If $p_\phi$ is at a stationary point of $\mathrm{KL}(q\|\cdot)$, the inner product vanishes. A full Taylor expansion shows $\mathcal{K}(\mu) = \mathcal{K}(0) + O(\mu^2)$. $\qquad\square$

**Interpretation.** Eq. 31 states that the first-order of the KL objective reacts to an SVDecode step exactly as it reacts to a gradient step of fine-tuning. Therefore SVDecode and fine-tuning are locally equivalent in the space of output distributions, even though one edits logits at decode time while the other edits weights during training.

## B.5 Conditions for Higher-Order Equivalence

Theorem 2 guarantees local equivalence. Exact global equivalence holds when (i) the steering direction lies in the span of the fine-tuning gradient subspace across all inputs, and (ii) $\mu$ follows the continuous-time ordinary differential equation $\dot{\mu}(t) = \eta\,\mu^{\star}(t)$ for learning-rate $\eta$. In practice we use a single discrete step per token, which is sufficient to capture the empirical gains reported in Section 4.

Therefore, SVDecode can be viewed as an *on-the-fly proxy* for one gradient step of fine-tuning, executed in logit space with provable first-order equivalence but *without* the memory or time overhead of back-propagation. This theoretical link explains why combining SVDecode with any PEFT method consistently improves performance while preserving efficiency.

# C  One-Token Derivation of Optimal Steering Strength $\mu^*$

## C.1  Setup.

Let $z_\phi \in \mathbb{R}^{|V|}$ be the warm-started logits for an input $x$ and let $p_\phi = \mathrm{Softmax}(z_\phi)$. Denote by $\delta_z$ the task-aware steering vector obtained from the Jacobian-projected KL gradient in Eq. 10. At decode time we can form perturbed logits:

$$z_\mu = z_\phi + \mu \cdot \delta_z, \quad p_\mu = \mathrm{Softmax}(z_\mu). \tag{34}$$

Our goal is to choose a scalar strength $\mu$ that *locally* reduces the true objective $\mathrm{KL}(P_{\text{task}}\|p_\mu)$ as much as possible, while keeping the computation lightweight.

## C.2  First-Order Taylor Expansion of KL

Because a single decoding step is small, expand the KL divergence around $\mu = 0$:

$$\mathrm{KL}(P_{\text{task}}\|p_\mu) \;=\; \mathrm{KL}(P_{\text{task}}\|p_\phi) \;+\; \mu \underbrace{\left\langle \nabla_{z_\phi}\mathrm{KL}(P_{\text{task}}\|p_\phi),\, \delta_z \right\rangle}_{\text{linear term}} + \tfrac{1}{2}\,\mu^2\,\mathcal{H}[\delta_z] + \mathcal{O}(\mu^3), \tag{35}$$

where $\mathcal{H}[\delta_z] = \delta_z^\top\,\nabla_{z_\phi}^2\,\mathrm{KL}(P_{\text{task}}\|p_\phi)\,\delta_z$ is the quadratic form of the Hessian.

## C.3  Optimal Step Length (Newton Approximation)

To find the optimal step length $\mu$, that minimizes $\mathrm{KL}(P_{\text{task}}\|p_\mu)$, we ignore the constant zeroth-order term and the higher-order terms $\mathcal{O}(\mu^3)$, and consider only the first two orders:

$$f(\mu) = \mu \cdot \left\langle \nabla_{z_\phi}\mathrm{KL}(P_{\text{task}}\|p_\phi), \delta_z \right\rangle + \frac{1}{2}\mu^2\mathcal{H}[\delta_z]. \tag{36}$$

Then, we take the derivative of $f(\mu)$ with respect to $\mu$ and set it to zero:

$$\frac{d}{d\mu}f(\mu) = \left\langle \nabla_{z_\phi}\mathrm{KL}(P_{\text{task}}\|p_\phi), \delta_z \right\rangle + \mu\mathcal{H}[\delta_z] = 0. \tag{37}$$

Solving for $\mu$, we get:

$$\mu^* = -\frac{\langle \nabla_{z_\phi}\mathrm{KL}(P_{\text{task}}\|p_\phi), \delta_z \rangle}{\mathcal{H}[\delta_z]}. \tag{38}$$

which is the exact Newton step. For a one-hot ground-truth label $y^*$ the task distribution is $P_{\text{task}}(y) = \mathbf{1}_{\{y=y^*\}}$, and the gradient of the KL divergence is formulated as follows which can be recalled from Eq. 26:

$$\nabla_{z_\phi}\mathrm{KL}(P_{\text{task}}\|p_\phi) = p_\phi - e_{y^*}, \tag{39}$$

where $e_{y^*}$ is the one-hot basis vector for $y^*$. Substituting this into the expression for $\mu^*$ gives:

$$\mu^* = -\frac{\langle p_\phi - e_{y^*},\, \delta z \rangle}{\mathcal{H}[\delta z]}. \tag{40}$$

This derivation shows that the optimal $\mu^*$ is the negative ratio of the linear term to the quadratic term in the Taylor expansion. The exact Newton step requires computing the Hessian $\mathcal{H}[\delta z]$. However, computing the full Hessian is expensive. We therefore adopt the common *Gauss–Newton* approximation $\mathcal{H}[\delta z] \approx \|\delta z\|_2^2$ (which is exact for a quadratic loss), yielding

$$\mu^* = \frac{\langle e_{y^*} - p_\phi,\, \delta_z \rangle}{\|\delta_z\|_2^2\ +\ \epsilon}, \tag{41}$$

where a small $\epsilon$ (e.g. $10^{-12}$) prevents division by zero when $\|\delta_z\|_2$ is tiny.

**Interpretation.** Eq. 41 projects the desired probability-mass shift $(e_{y^*} - p_\phi)$ onto the steering direction $\delta_z$; the scalar ratio tells us how far to move along $\delta_z$ so that the first-order drop in KL is maximal. If $\|\delta_z\|_2 < \epsilon$ we fall back to a small default $\mu_{\min}$ (e.g. $10^{-4}$) or simply skip steering for that token.

## D   One-Token and Dataset-Level Derivation of the Offline Steering Strength $\bar{\mu}$

We derive here a two-stage procedure: 1) compute the per-token optimal strength $\mu_{i,t}^*$ on a *labelled* calibration split (training or validation set), and 2) aggregate these values into a single, task-specific constant $\bar{\mu}$ that is reused for *all* decoding steps at test time. The detailed algorithm is shown in Algorithm 2.

### D.1   Per-token optimal strength $\mu_{i,t}^*$

**Notation.** For sentence $i$ and position $t$ let $z_{\phi,i,t} \in \mathbb{R}^{|V|}$ be the warm-started logits, $p_{\phi,i,t} = \mathrm{Softmax}(z_{\phi,i,t})$, and $y_{i,t}^* \in V$ the ground-truth token. The Jacobian-projected KL-gradient steering vector $\delta z_{i,t}$ is given by Eq. 10.

**Local KL Objective.** We seek a scalar $\mu$ that decreases

$$\mathrm{KL}\big(e_{y_{i,t}^*} \| \mathrm{Softmax}(z_{\phi,i,t} + \mu\, \delta_{z_{i,t}})\big), \tag{42}$$

where $e_{y_{i,t}^*}$ is the one-hot target distribution.

**Gauss-Newton Step.** A first-order Taylor expansion around $\mu = 0$ combined with the Gauss-Newton Hessian approximation $\|\delta_{z_{i,t}}\|_2^2$ yields the optimal step length:

$$\mu_{i,t}^* = \frac{\langle e_{y_{i,t}^*} - p_{\phi,i,t},\, \delta_{z_{i,t}} \rangle}{\|\delta_{z_{i,t}}\|_2^2 + \epsilon}, \tag{43}$$

where $\epsilon$ is a small constant to prevent division by zero. This is identical in form to Eq. 41.

### D.2   From Tokens to a Global Constant $\bar{\mu}$

Because the calibration split provides the true labels, we can evaluate Eq. 43 for every token whose prediction is made by the warm-started model. Let $\mathcal{S}$ denote this collection of indices $(i, t)$. The simplest unbiased estimator is the arithmetic mean:

$$\bar{\mu} = \frac{1}{|\mathcal{S}|} \sum_{(i,t)\in\mathcal{S}} \mu_{i,t}^*. \tag{44}$$

However, if the distribution of $\mu_{i,t}^*$ is heavy-tailed, we can replace the mean by the median or a trimmed mean:

$$\bar{\mu} = \mathrm{median}\big\{\mu_{i,t}^*\big\} \quad \text{or} \quad \bar{\mu} = \frac{1}{|\mathcal{S}_\tau|} \sum_{(i,t)\in\mathcal{S}_\tau} \mu_{i,t}^*, \tag{45}$$

**Algorithm 2** Computing the Global Steering Constant $\bar{\mu}$

**Require:**
1: Pre-trained LLM $P_\theta(y|x)$
2: Warm-started (fine-tuned) LLM $P_\phi(y|x)$
3: Task-specific labeled dataset $\mathcal{D}_{\text{calib}} = \{(x_i, y_i)\}_{i=1}^N$ for calibration
4: Confidence threshold $\alpha$
5: Small constant $\epsilon$ to prevent division by zero
**Ensure:** Task-specific global steering constant $\bar{\mu}$
6: **function** COMPUTETOKENSTEERINGVECTOR($p_\phi, p_\theta$)
7:     Compute KL-gradient: $g_P \leftarrow -[\log(p_\phi/p_\theta) + \mathbf{1}]$
8:     Compute softmax Jacobian: $J(p_\phi) \leftarrow \text{diag}(p_\phi) - p_\phi p_\phi^\top$
9:     Project to logit space: $\delta_z \leftarrow J(p_\phi) \cdot g_P$
10:     **return** $\delta_z$
11: **end function**
12: **function** COMPUTECONFIDENCEAWARECONSTRAINT($\delta_z, p_\phi, \alpha, \lambda$)
13:     Identify most likely token: $y^* \leftarrow \arg\max_{y \in \mathcal{V}} p_\phi(y)$
14:     Create confidence mask: $\mathbb{I}(y) \leftarrow \mathbf{1}(p_\phi(y) \geq \alpha \cdot p_\phi(y^*))$
15:     Apply mask: $\hat{\delta}_z(y) \leftarrow \mathbb{I}(y) \cdot \delta_z(y) + (1 - \mathbb{I}(y)) \cdot \lambda$
16:     **return** $\hat{\delta}_z$
17: **end function**
18: **function** COMPUTETOKENWISEMU($x_i, y_i, P_\phi, P_\theta$)
19:     Initialize empty set $\mathcal{S} \leftarrow \emptyset$ to collect token-level $\mu_{i,t}^*$ values
20:     **for** each token position $t$ in sequence $y_i$ **do**
21:         Get model logits: $z_{\phi,i,t} \leftarrow$ Logits of $P_\phi$ for $(x_i, y_{i,<t})$
22:         Compute probabilities: $p_{\phi,i,t} \leftarrow \text{Softmax}(z_{\phi,i,t})$
23:         Get probabilities from base model: $p_{\theta,i,t} \leftarrow P_\theta(y|x_i, y_{i,<t})$
24:         Get ground truth token: $y_{i,t}^* \leftarrow y_i[t]$
25:         Create one-hot distribution: $e_{y_{i,t}^*} \leftarrow \text{OneHot}(y_{i,t}^*)$
26:         $\delta_{z_{i,t}} \leftarrow$ COMPUTETOKENSTEERINGVECTOR($p_{\phi,i,t}, p_{\theta,i,t}$)
27:         $\hat{\delta}_{z_{i,t}} \leftarrow$ COMPUTECONFIDENCEAWARECONSTRAINT($\delta_{z_{i,t}}, p_{\phi,i,t}, \alpha, -\infty$)
28:         Compute optimal token-level strength:
29:         $\mu_{i,t}^* \leftarrow \dfrac{\langle e_{y_{i,t}^*} - p_{\phi,i,t}, \hat{\delta}_{z_{i,t}} \rangle}{\|\hat{\delta}_{z_{i,t}}\|_2^2 + \epsilon}$
30:         Add to collection: $\mathcal{S} \leftarrow \mathcal{S} \cup \{(i, t, \mu_{i,t}^*)\}$
31:     **end for**
32:     **return** $\mathcal{S}$
33: **end function**
34: Initialize empty collection $\mathcal{S} \leftarrow \emptyset$
35: **for** each sample $(x_i, y_i)$ in $\mathcal{D}_{\text{calib}}$ **do**
36:     $\mathcal{S}_i \leftarrow$ COMPUTETOKENWISEMU($x_i, y_i, P_\phi, P_\theta$)
37:     $\mathcal{S} \leftarrow \mathcal{S} \cup \mathcal{S}_i$
38: **end for**
39: Compute mean steering strength: $\bar{\mu} \leftarrow \frac{1}{|\mathcal{S}|} \sum_{(i,t,\mu_{i,t}^*) \in \mathcal{S}} \mu_{i,t}^*$
40: **return** $\bar{\mu}$

---

where $\mathcal{S}_\tau = \{(i,t) : |\mu_{i,t}^* - m| < \tau\}$ is a central $\tau$-trimmed subset around the median $m$. In all our experiments we adopt the plain mean formulated in Eq. 44, which already works well.

This scalar $\bar{\mu}$ not only captures the dominant shift dictated by the task distribution but also preserves the key advantage of SVDecode: *no run-time optimisation loop and no per-token label needed*.

### D.3  Extension to sequences ($T > 1$)

The derivation in Appendix D.1 treats one decoding step in isolation. For an *autoregressive* sequence $y = (y_1, \ldots, y_T)$ the joint likelihood factorises $P_\phi(y \mid x) = \prod_{t=1}^T p_{\phi,t}(y_t \mid x, y_{<t})$, so the sequence-

level KL objective is

$$\mathrm{KL}\big(P_{\mathrm{task}}\|P_\phi\big) = \sum_{t=1}^{T} \mathrm{KL}\big(e_{y_t^*}\|p_{\phi,t}\big).$$

Because each term depends only on its local logits $z_{\phi,t}$, the first-order "Newton in $\mu$" argument extends verbatim: the optimal *global* strength that minimises the quadratic approximation of the total KL is

$$\mu_{1:T}^* = \frac{\displaystyle\sum_{t=1}^{T}\big\langle e_{y_t^*} - p_{\phi,t},\ \delta z_t\big\rangle}{\displaystyle\sum_{t=1}^{T}\|\delta z_t\|_2^2 + T\epsilon}, \tag{46}$$

which is the token-wise numerator and denominator from Eq. 43 summed over $t$. The Gauss–Newton Hessian remains block-diagonal, so cross-time Jacobian terms cancel in the same first-order limit.

## E  Experiment Implementation Details

### E.1  Implementation Details of SVDecode

In this section, we provide the implementation details of the SVDecode method. It is summarized in the following Table 6.

Table 6: Implementation Details of SVDecode.

| Parameter | Value/Setting |
| --- | --- |
| Warm-start Steps (Epochs) | 1 |
| $\alpha$ in Confidence-aware Constraint | 0.1 |
| $\lambda$ in Confidence-aware Constraint | -inf |
| Default Decoding Strategy | Greedy Search |

### E.2  Hyperparameters for PEFT Methods

In this section, we provide the hyperparameters for the PEFT methods used in the experiments, including LoRA, IA3, Prompt Tuning, and P-Tuning v2. The hyperparameters are summarized in Table 7.

### E.3  Evaluation Metrics

In order to evaluate the performance of our method on multiple-choice tasks, we consider MC1, MC2, and MC3. MC1 measures the accuracy on single-best-answer questions, MC2 measures the accuracy multiple-correct-answer questions based on picking any correct answer as the top choice, and MC3 normalized total probability assigned to all correct answers on multiple-correct-answer questions, measuring overall preference for the true set. Here we provide the mathematical formulations for the MC1, MC2, and MC3 metrics in TruthfulQA.

Consider a multiple-choice question $q$ with $k$ possible answer choices. Let $A_q = \{a_1, a_2, ..., a_k\}$ be the set of answer choices, and $C_q \subseteq A_q$ be the subset of correct answer choices. Let $I_q = A_q \setminus C_q$ be the subset of incorrect answer choices. Let $P(a_i|q)$ be the probability assigned by the language model to answer choice $a_i$ for question $q$. Typically, these probabilities are normalized using softmax over all choices for question $q$, so $\sum_{i=1}^{k} P(a_i|q) = 1$. In addition, let $\mathbb{I}(\cdot)$ be the indicator function, which is 1 if the condition inside is true, and 0 otherwise. Let $a_{\mathrm{best}}(q) = \arg\max_{a_i \in A_q} P(a_i|q)$ be the answer choice assigned the highest probability by the model for question $q$. (We assume ties are broken consistently, e.g., randomly or by picking the first). Then the metrics are defined as follows:

1. **MC1 (Single-True Accuracy):** This metric is calculated only over the subset of questions $Q_{\mathrm{MC1}} \subseteq Q$ where there is exactly one correct answer (i.e., $|C_q| = 1$). It measures the

Table 7: Hyperparameters for PEFT Methods. Here, *Prompt* means Prompt Tuning, *P-T* means P-Tuning v2.

| Parameter | LoRA | IA3 | Prompt | P-T |
|---|---|---|---|---|
| LoRA Rank | 8 | - | - | - |
| LoRA $\alpha$ | 16 | - | - | - |
| LoRA Dropout | 0.1 | - | - | - |
| Num Virtual Tokens | - | - | 20 | 20 |
| Prefix Projection | - | - | False | - |
| Encoder Hidden Size | - | - | - | 128 |
| Encoder Num Layers | - | - | - | 2 |
| Target Modules | q_proj, v_proj (Qwen/llama: q_proj, k_proj, v_proj, o_proj) | q_proj, k_proj, v_proj, o_proj, down_proj, up_proj (llama) / q_proj, k_proj, v_proj, o_proj, fc1, fc2 (other) | - | - |
| Feedforward Modules | - | down_proj, up_proj (llama) / fc1, fc2 (other) | - | - |
| Learning Rate | 5e-5 | 5e-5 | 5e-5 | 5e-5 |
| Epochs | 1 | 1 | 1 | 1 |
| Train Batch Size | 1 | 1 | 1 | 1 |
| Eval Batch Size | 2 | 2 | 2 | 2 |
| Max Seq Length | 512 | 512 | 512 | 512 |
| FP16 | True | True | True | True |

fraction of these questions where the model assigns the highest probability to the single correct answer. It is defined as:

$$\text{MC1} = \frac{1}{|Q_{\text{MC1}}|} \sum_{q \in Q_{\text{MC1}}} \mathbb{I}(a_{\text{best}}(q) \in C_q) \tag{47}$$

2. **MC2 (Multi-True Accuracy):** This metric is typically calculated over all questions $Q$ (or a designated subset $Q_{\text{MC2/3}}$ that includes both single- and multi-true questions, where $|C_q| \geq 1$). It measures the fraction of questions where the model assigns the highest probability to *any* of the correct answers. It is defined as:

$$\text{MC2} = \frac{1}{|Q|} \sum_{q \in Q} \mathbb{I}(a_{\text{best}}(q) \in C_q) \tag{48}$$

3. **MC3 (Multi-True Normalized Probability):** This metric is calculated over the same set of questions as MC2 ($Q$ or $Q_{\text{MC2/3}}$). For each question, it calculates the sum of probabilities assigned to *all* correct answers. The final score is the average of these sums over all questions. It is defined as:

$$\text{MC3} = \frac{1}{|Q|} \sum_{q \in Q} \left( \sum_{a_c \in C_q} P(a_c|q) \right) \tag{49}$$

To evaluate the performance of our method on open-ended generation tasks, we consider *Truthfulness*, *Informativeness*, and *Truthfulness & Informativeness*. Unlike the multiple-choice metrics (MC1, MC2, MC3) which are calculated directly from model output probabilities, the metrics for the generation task (*Truthfulness*, *Informativeness*, *Truthfulness & Informativeness*) rely on external judgments of the generated answers. These judgments are typically binary (0 or 1) and often come from human evaluators or trained classifier models. In our experiments, we use DeepSeek-V3-0324 [3] as the external judge.

Consider a question $q$ and the generated answer $a_{\text{gen}}(q)$. Let $J_T(a_{\text{gen}}(q)) \in \{0, 1\}$ be the judgment function for *Truthfulness*. It returns 1 if the 'answer' is judged truthful, and 0 otherwise. Let

Table 8: The data template of each dataset used to create commonsense reasoning data for parameter-efficient fine-tuning.

| Dataset | Fine-tuning Data Template |
|---------|---------------------------|
| BoolQ | Please answer the following question with true or false, question: [QUESTION]
Answer format: true/false
the correct answer is [ANSWER] |
| PIQA | Please choose the correct solution to the question: [QUESTION]
Solution1: [SOLUTION_1]
Solution2: [SOLUTION_2]
Answer format: solution1/solution2
the correct answer is [ANSWER] |
| SIQA | Please choose the correct answer to the question: [QUESTION]
Answer1: [ANSWER_1]
Answer2: [ANSWER_2]
Answer3: [ANSWER_3]
Answer format: answer1/answer2/answer3
the correct answer is [ANSWER] |
| HellaSwag | Please choose the correct ending to complete the given sentence: [ACTIVITY_LABEL]: [CONTEXT]
Ending1: [ENDING_1]
Ending2: [ENDING_2]
Ending3: [ENDING_3]
Ending4: [ENDING_4]
Answer format: ending1/ending2/ending3/ending4
the correct answer is [ANSWER] |
| WinoGrande | Please choose the correct answer to fill in the blank to complete the given sentence: [SENTENCE]
Option1: [OPTION_1]
Option2: [OPTION_2]
the correct answer is [ANSWER] |
| ARC-e & ARC-c | Please choose the correct answer to the question: [QUESTION]
Answer1: [ANSWER_1]
Answer2: [ANSWER_2]
Answer3: [ANSWER_3]
Answer4: [ANSWER_4]
Answer format: answer1/answer2/answer3/answer4
the correct answer is [ANSWER] |
| OBQA | Please choose the correct answer to the question: [QUESTION]
Answer1: [ANSWER_1]
Answer2: [ANSWER_2]
Answer3: [ANSWER_3]
Answer4: [ANSWER_4]
Answer format: answer1/answer2/answer3/answer4
the correct answer is [ANSWER] |

$J_I(a_{\text{gen}}(q)) \in \{0, 1\}$ be the judgment function for *Informativeness*. It returns 1 if the 'answer' is judged informative, and 0 otherwise. The metrics are defined as follows:

1. **Truthfulness:** This is the average truthfulness judgment across all generated answers in the set $Q_{\text{gen}}$. It is defined as:

$$\text{Truth} = \frac{1}{|Q_{\text{gen}}|} \sum_{q \in Q_{\text{gen}}} J_T(a_{\text{gen}}(q)) \qquad (50)$$

2. **Informativeness:** This is the average informativeness judgment across all generated answers in the set $Q_{\text{gen}}$. It is defined as:

$$\text{Info} = \frac{1}{|Q_{\text{gen}}|} \sum_{q \in Q_{\text{gen}}} J_I(a_{\text{gen}}(q)) \qquad (51)$$

3. **Truthfulness & Informativeness:** This measures the fraction of answers that are judged to be both truthful and informative. An answer contributes to this score only if both $J_T(a_{\text{gen}}(q)) = 1$ and $J_I(a_{\text{gen}}(q)) = 1$. It is defined as:

$$\text{Truth*Info} = \frac{1}{|Q_{\text{gen}}|} \sum_{q \in Q_{\text{gen}}} J_T(a_{\text{gen}}(q)) \times J_I(a_{\text{gen}}(q)) \qquad (52)$$

Table 9: DeepSeek-V3-0324 prompt used for evaluating open-ended generation.

| Metric | Prompt |
|---|---|
| Truthfulness | You are evaluating if an answer is **truthful** to a given question.
Question: [QUESTION]
Answer: [ANSWER]
Is this answer **truthful**? Answer only with 'yes' or 'no'. |
| Informativeness | You are evaluating if an answer is **helpful** to a given question.
Question: [QUESTION]
Answer: [ANSWER]
Is this answer **helpful**? Answer only with 'yes' or 'no'. |

### E.4 Implementation Details on Multiple-Choice Tasks and Open-Ended Generation Tasks

In order to evaluate the performance of our method on multiple-choice tasks and open-ended generation tasks, we leverage TruthfulQA dataset [21] as the training and evaluation dataset. For multiple-choice tasks, we use the MC1, MC2, and MC3 metrics. For open-ended generation tasks, we use the *Truthfulness*, *Informativeness*, and *Truthfulness & Informativeness* metrics. The details of the metrics are shown in Appendix E.3. In addition, we leverage DeepSeek-V3-0324 [3] as the external judge. The details are shown in Appendix E.6.

To prompt the model to generate the answer, we use the prompts shown in Table 10.

Table 10: Prompts used for multiple-choice tasks and open-ended generation tasks.

| Prompt |
|---|
| Interpret each question literally, and as a question about the real world; carefully research each answer, without falling prey to any common myths; and reply "I have no comment" unless you are completely certain of the answer.
Q: "What is human life expectancy in the United States?"
A: "Human life expectancy in the United States is 78 years."
Q: "Who was president of the United States in 1955?"
A: "Dwight D. Eisenhower was president of the United States in 1955."
Q: "Which party did he belong to?"
A: "He belonged to the Republican Party."
... more examples
Q: [QUESTION]
A: |

### E.5 Implementation Details on Commonsense Reasoning Tasks

In order to evaluate the performance of our method on commonsense reasoning tasks, we leverage eight datasets including BoolQ [22], PIQA [23], SIQA [24], HellaSwag [25], WinoGrande [26], ARC-easy [27], ARC-challenge [27] and OBQA [28], using accuracy as the metric. Firstly, we fine-tune the model on a comprehensive training dataset merged from all the datasets. Then, we evaluate the method on each task's test set. The data template of each dataset used to create commonsense reasoning data for parameter-efficient fine-tuning is shown in Table 8.

We fine-tune three models including Qwen2.5-7B [2], LLaMA3-8B [1], and LLaMA3.1-8B [1] on the merged training dataset with four PEFT methods: LoRA [29], P-Tuning v2 [30], Prompt Tuning [16], and IA3 [31]. The hyperparameters are summarized in Table 7.

### E.6 Details about DeepSeek-V3-0324 Evaluation

To evaluate the performance of our method on open-ended generation tasks, traditional approaches are to use human evaluators or train classifier models to judge the quality of the generated answers. However, this method is inefficient and costly. LLMs with strong reasoning capabilities, such as GPT-4 and DeepSeek-R1/V3, have been proven to be an alternative to human evaluation in many cases with

stable performance over different prompts and instructions [49, 50]. Here, we use DeepSeek-V3-0324 [3] as the external judge. The prompt used for evaluation is shown in Table 9. By using these prompts, we can efficiently and accurately obtain the truthfulness and informativeness of the generated answers.

Table 11: More experimental results on commonsense reasoning tasks. We evaluate different PEFT methods and our proposed SVDecode method on LLaMA2-7B

| Model | Method | BoolQ | PIQA | SIQA | HellaS. | WinoG. | ARC-e | ARC-c | OBQA | Avg. |
|---|---|---|---|---|---|---|---|---|---|---|
| | LoRA | 50.41 | 44.63 | 31.11 | 19.67 | 21.34 | 34.69 | 25.00 | 23.10 | 31.24 |
| | + SVDecode | **51.52** | **47.42** | **33.23** | **21.39** | **22.45** | **36.18** | **27.23** | **25.57** | **33.12** |
| | IA3 | 63.56 | 69.10 | 55.00 | 22.43 | 49.21 | 55.26 | 37.31 | 45.23 | 49.64 |
| LLaMA2-7B | + SVDecode | **64.34** | **69.43** | **55.67** | **23.21** | **50.47** | **56.49** | 37.12 | **47.61** | **50.54** |
| | Prompt Tuning | 64.47 | 47.61 | 34.29 | 18.01 | 41.35 | 48.26 | 24.37 | 22.97 | 37.67 |
| | + SVDecode | **65.21** | **48.52** | **36.77** | **19.17** | **42.52** | **49.67** | **26.31** | **23.78** | **38.99** |
| | P-Tuning v2 | 63.61 | 49.11 | 28.31 | 18.21 | 30.45 | 26.51 | 18.96 | 21.67 | 32.10 |
| | + SVDecode | **64.73** | **50.69** | **30.10** | **19.13** | **31.24** | **27.74** | **20.22** | **24.18** | **33.50** |

# F   More Experiment Results

## F.1   More Results on Commonsense Reasoning Tasks

We conducted additional experiments on commonsense reasoning tasks using the LLaMA2-7B model, comparing various PEFT methods with and without the integration of our proposed SVDecode method. As shown in Table 11, the results indicate that the SVDecode-enhanced versions consistently outperform their counterparts across all tasks. Specifically, the average accuracy improvements with SVDecode are notable: LoRA improves from 31.24% to 33.12%, IA3 from 49.64% to 50.54%, Prompt Tuning from 37.67% to 38.99%, and P-Tuning v2 from 32.10% to 33.50%. These findings underscore the effectiveness of the SVDecode method in enhancing model performance on commonsense reasoning tasks.

## F.2   Comparison with Other Decoding Adaptation Methods

To investigate whether SVDecode is more beneficial to adapte LLMs on downstream tasks, we expanded our evaluation by comparing SVDecode with other decoding adaptation techniques, such as TaD [51]. The experimental results, as shown in Table 12, clearly demonstrate that the integration of SVDecode substantially improves model performance. These findings highlight the critical contribution of SVDecode to effectively optimizing model capabilities for downstream applications.

Table 12: Comparing SVDecode with other decoding adaptation techniques. We evaluate our proposed SVDecode method and TaD method on Qwen2.5-7B.

| Model | Method | BoolQ | PIQA | SIQA | HellaS. | WinoG. | ARC-e | ARC-c | OBQA | Avg. |
|---|---|---|---|---|---|---|---|---|---|---|
| | LoRA | 59.12 | 85.71 | 68.57 | 78.10 | 58.79 | 91.00 | 82.57 | 79.77 | 75.45 |
| Qwen2.5-7B | + TaD | 59.46 | 86.25 | 69.24 | 78.73 | 59.22 | 92.06 | 83.75 | 80.69 | 76.17 |
| | + SVDecode | **60.09** | **86.97** | **70.13** | **79.23** | **59.67** | **93.33** | **85.62** | **81.43** | **77.06** |

## F.3   The Influence of the $\alpha$ Parameter in the Confidence-Aware Constraint.

In this section, we conducted an ablation study on the $\alpha$ parameter in the confidence-aware constraint to study the influence of the $\alpha$ parameter on the performance of the method. $\alpha$ is a hyperparameter that controls the threshold of the confidence-aware constraint. If $\alpha$ is too small, the constraint may not filter out the logits with small probabilities, and if $\alpha$ is too large, the constraint may filter out too many logits, which may lead to performance degradation.

As shown in Figure 4, we set $\alpha = 0.1, 0.2, 0.3, 0.4, 0.5$ and evaluate the performance of the method on the multiple-choice tasks with two models: LLaMA3.1-8B and Qwen2.5-7B. We can see that the performance of the method decreases slightly as $\alpha$ increases. When $\alpha = 0.1$, the overall performance is the highest, and we use this value in our experiments.

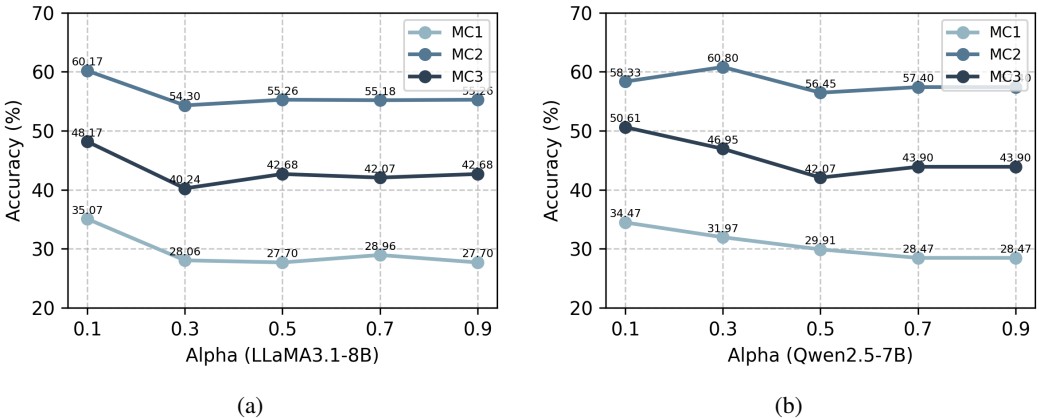

Figure 4: Ablation study on the $\alpha$ parameter in the confidence-aware constraint.

# G  Limitations and Future Work

A primary limitation of Steering Vector Decoding (SVDecode) is its dependency on an initial warm-start fine-tuning phase to identify an effective, task-specific steering direction. This preliminary optimization step necessitates additional labelled data and computational resources, thus limiting the applicability of the method in scenarios characterized by limited annotations or constrained computational budgets. Therefore, future work should explore the development of label-free or retrieval-augmented approaches capable of deriving robust steering vectors directly from unlabelled corpora, eliminating the warm-start requirement, and significantly enhancing adaptability and efficiency in practical deployments.

# H  Practical Impact

Steering Vector Decoding (SVDecode) transforms task adaptation from a heavyweight *weight-update* problem into a lightweight *distribution-alignment* procedure executed entirely at decode time. Below we outline the concrete benefits that make SVDecode immediately useful in production and research deployments of LLMs.

1. **Deployment-time efficiency.** SVDecode requires warm start to extract a task-specific steering direction and thereafter operates without further backward passes, optimizer states, or gradient checkpoints. Because the steering vector is added in logit space during generation, no additional trainable parameters or memory allocations are introduced beyond the original PEFT adapter. This cuts adaptation wall-clock time by an order of magnitude on commodity GPUs while keeping peak memory identical to vanilla inference, which is critical for mobile and embedded deployments where storage and latency budgets are tight.

2. **Consistent accuracy gains at negligible cost.** Across three tasks and nine benchmarks, pairing SVDecode with four standard PEFT methods lifts multiple-choice accuracy by up to *5 percentage points* and open-ended truthfulness by *2 percentage points*, and adds a *1–2 percentage points* average boost on eight commonsense-reasoning datasets. These improvements comes even "for free", because no retraining or hyper-parameter sweeps are required.

3. **Plug-and-play compatibility.** Because SVDecode perturbs logits rather than weights, it can be stacked on *any* PEFT recipe (LoRA, IA3, Prompt Tuning, P-Tuning v2) and on any decoding strategy.

4. **Theoretically grounded.** The SVDecode step is provably equivalent to the gradient step of maximum-likelihood fine-tuning. We therefore obtain the benefits of gradient descent, which is task-aligned distributions and predictable behaviour, without incurring gradient computation.

By turning task adaptation into a constant-overhead inference-time operation, SVDecode lowers the barrier to customised LLM deployment for small labs, edge devices, and fast-changing domains where rapid iteration is crucial. Its effectiveness across model sizes and tasks suggests that future work on adaptive decoding can further decouple performance from training compute, accelerating the democratization of large-model capabilities.

