# OpenReview forum: "Distribution-Aligned Decoding for Efficient LLM Task Adaptation"
_NeurIPS.cc/2025/Conference — NeurIPS 2025 poster_

### Official Review · Reviewer_kREE · 2025-06-13

**Clarity:** 3
**Significance:** 4
**Originality:** 4
**Rating:** 6
**Confidence:** 4

**Summary:**

The paper "Distribution-Aligned Decoding for Efficient LLM Task Adaptation" proposes a novel approach to adapt large language models (LLMs) for downstream tasks by aligning the model's output distribution with the task-specific distribution during decoding, rather than relying solely on weight updates. The authors introduce Steering Vector Decoding (SVD), a lightweight and theoretically grounded method compatible with parameter-efficient fine-tuning (PEFT) techniques.
SVD operates in two steps:
- Steering Vector Construction: After a brief warm-start fine-tuning phase, a task-aware steering vector is derived from the gradient of the Kullback-Leibler (KL) divergence between the warm-started and pre-trained model output distributions. This vector is projected into logit space and refined with a confidence-aware constraint to ensure stability and relevance.
- Task-Aware Decoding: The steering vector adjusts the model's logits during decoding to align the output distribution with the task, using an optimal steering strength derived theoretically.
The method is proven to be equivalent to a gradient step in full fine-tuning, offering a computationally efficient alternative. Evaluated across three tasks and nine benchmarks, SVD enhances performance when paired with four PEFT methods (LoRA, IA3, Prompt Tuning, P-Tuning v2), improving multiple-choice accuracy by up to 5 points, open-ended truthfulness by 2 points, and commonsense reasoning by 1-2 points, without adding trainable parameters beyond the PEFT adapter.

**Questions:**

- Can you provide formal statistical analysis (e.g., error bars, confidence intervals, or significance tests) for the experimental results, particularly for the commonsense reasoning tasks where gains are modest (1-2 points)?

- Why was the comparison of SVD limited to TaD, and can you include evaluations against other inference-time adaptation methods (e.g., contrastive decoding, activation steering)?

- How does SVD perform on larger models (e.g., 70B parameters) or more diverse tasks (e.g., multilingual or domain-specific datasets), and can you provide such evaluations?

- Can you elaborate on potential failure modes of SVD, particularly when the warm-start distribution is poorly aligned with the task, and discuss mitigation strategies? A suggestion: expand the “Limitations and Future Work” section to include 1-2 specific failure scenarios (e.g., noisy warm-start data) and propose mitigations (e.g., adaptive steering strength).

- Can you provide detailed computational resource requirements (e.g., execution time, memory usage) for SVD’s warm-start and decoding phases to aid reproducibility? A suggestion: add a table or appendix specifying GPU type, memory, and runtime for each experiment (e.g., for LLaMA2-7B with LoRA+SVD).

**Ethical Concerns:**

["NO or VERY MINOR ethics concerns only"]

**Final Justification:**

Very strong paper. I stand by the comment and the score given the rebuttal and the comments made by other reviewers.

**Limitations:**

yes

**Paper Formatting Concerns:**

No.

**Quality:**

4

**Strengths And Weaknesses:**

*Quality*
Technical Soundness: The paper is theoretically grounded, with a clear derivation of the equivalence between the negative log-likelihood (NLL) objective and minimizing the expected Kullback-Leibler (KL) divergence (Theorem 1, Page 3). The proof is rigorous, correctly leveraging the delta function properties of the empirical label distribution, and is well-supported by mathematical formalism:
$$\mathcal{L}_{\mathrm{FT}}(\theta) = \mathbb{E}_{x \sim \mathcal{D}}\left[\mathrm{KL}\left(\hat{P}_{\text{task}}(y \mid x) \| P_\theta(y \mid x)\right)\right]$$
The authors further prove that Steering Vector Decoding (SVD) is first-order equivalent to a gradient step in full fine-tuning, providing a strong theoretical foundation for the method.
Robust Experimental Evaluation: The paper evaluates SVD across three tasks (multiple-choice, open-ended generation, and commonsense reasoning) and nine benchmarks (e.g., TruthfulQA, BoolQ, PIQA), using multiple models (LLaMA2-7B, Qwen2.5-7B, LLaMA3-8B, LLaMA3.1-8B) and four PEFT methods (LoRA, IA3, Prompt Tuning, P-Tuning v2). Results are consistent, with improvements of up to 5 points in multiple-choice accuracy, 2 points in open-ended truthfulness, and 1-2 points in commonsense reasoning (Table 11, Page 26).
Ablation Studies: The ablation study on the $\alpha$ parameter (Figure 4, Page 27) demonstrates the impact of the confidence-aware constraint, showing that $\alpha = 0.1$ yields optimal performance. This adds credibility to the method’s design choices.
Limited Statistical Analysis: While the paper reports results averaged across benchmarks and includes ablations, it lacks explicit error bars or confidence intervals for most experiments. The justification for statistical significance (Page 31) relies on “consistent deltas” across benchmarks and ablations, which is indirect and less rigorous than formal statistical tests (e.g., t-tests or bootstrap methods). This could weaken confidence in the generalizability of the results, especially for smaller datasets.
Comparison to Baselines: The comparison with other decoding adaptation methods is limited to TaD (Table 12, Page 26), which is insufficient to fully contextualize SVD’s performance. Including more baselines, such as contrastive decoding or other inference-time adaptation techniques, would strengthen the claim of superiority.
Computational Resource Details: While the paper claims to provide sufficient information on compute resources (Page 32), it lacks specifics on execution time or memory usage for individual experiments, which could hinder reproducibility for researchers with limited hardware.

*Clarity*
Clear Problem Statement: The introduction (Page 1) effectively motivates the problem of costly LLM adaptation, highlighting the resource constraints of full fine-tuning (e.g., LLaMA-7B requiring 58 GB of memory) and positioning PEFT as a partial solution. The reframing of task adaptation as output-distribution alignment is intuitive and well-articulated.
Structured Methodology: The method section (Pages 3-5) is logically organized, with subsections on steering vector construction (warm-start, KL gradient, logit-space projection, confidence-aware constraint) and task-aware decoding. Figure 1 (Page 4) visually clarifies the two-step SVD process, enhancing understanding.
Accessible Mathematical Derivations: The derivations, such as the KL gradient computation (Page 4) and logit-space projection (Page 5), are detailed yet accessible, with clear explanations of each term:
$$\delta_{\text{logits}} = \left(\operatorname{diag}(P_\phi) - P_\phi P_\phi^{\top}\right) \cdot \left(-\log \frac{P_\phi}{P_\theta} - 1\right)$$
The authors explain the practical implications (e.g., normalization constraints, numerical stability) alongside the math, making it approachable for readers familiar with LLMs.
Well-Presented Results: Tables 11 and 12 (Page 26) clearly compare SVD-enhanced PEFT methods against baselines, with consistent formatting and metrics (accuracy, truthfulness). The ablation study (Figure 4, Page 27) is visually clear, showing the impact of $\alpha$ on performance.
Overuse of Jargon in Places: Terms like “simplex geometry violation” (Page 5) and “confidence-aware steering vector constraint” (Page 3) are introduced without sufficient explanation for readers less familiar with probabilistic modeling or LLM decoding. A brief glossary or more intuitive descriptions could improve accessibility.

*Significance*
Practical Impact: SVD addresses a critical bottleneck in LLM deployment by reducing adaptation costs without sacrificing performance. The claimed benefits—up to 5-point gains in multiple-choice accuracy, 2-point gains in truthfulness, and 1-2-point gains in commonsense reasoning (Page 27)—are substantial for resource-constrained settings, such as mobile or edge devices.
Broad Applicability: The method’s compatibility with any PEFT recipe (LoRA, IA3, Prompt Tuning, P-Tuning v2) and decoding strategy (Page 27) makes it a versatile tool for the NLP community. This plug-and-play nature enhances its adoption potential across diverse tasks and models.
Theoretical Contribution: Reframing task adaptation as output-distribution alignment and proving SVD’s equivalence to a gradient step (Page 3) offers a new perspective on fine-tuning. This could inspire further research into inference-time adaptation methods, reducing reliance on expensive weight updates.
Incremental Gains in Some Cases: While the 5-point gain in multiple-choice accuracy is impressive, the 1-2-point gains in commonsense reasoning (Table 11, Page 26) are modest. For tasks where PEFT methods already perform well (e.g., IA3 with 49.64% average accuracy), the added benefit of SVD (to 50.54%) may not justify the additional complexity in some applications.
Limited Novel Task Enablement: The paper focuses on improving existing tasks (e.g., multiple-choice, commonsense reasoning) rather than enabling new tasks or domains. Its significance would be greater if it demonstrated applicability to emerging areas, such as low-resource languages or multimodal tasks.

*Originality*
Novel Perspective: Reframing task adaptation as output-distribution alignment (Page 1) is a fresh conceptual contribution. This perspective shifts the focus from weight updates to inference-time distribution steering, which is underexplored in the LLM adaptation literature.
Innovative Method: SVD is a novel method that combines KL divergence gradients, logit-space projection, and confidence-aware constraints (Pages 3-5). Unlike traditional PEFT methods that modify weights, SVD operates entirely at decode time, offering a new paradigm for adaptation:
$$\hat{z}_\phi(y) = z_\phi(y) + \mu \cdot \left(\mathbb{I}(y) \cdot \delta_{\text{logits}}(y) + (1 - \mathbb{I}(y)) \cdot \lambda\right)$$
Theoretical Grounding: The proof of equivalence to a gradient step in full fine-tuning (Page 3) is a unique contribution, bridging inference-time methods with traditional optimization. This distinguishes SVD from heuristic-based decoding methods like contrastive decoding.

---

> ### Author Rebuttal · Authors · 2025-07-30
>
> Dear Reviewer kREE,
>
> Thanks for your valuable comments and the time you dedicated to reviewing this work. Here we carefully and elaborately reply to your concerns.
>
> **Q1: Can you provide formal statistical analysis (e.g., error bars, confidence intervals, or significance tests) for the experimental results, particularly for the commonsense reasoning tasks where gains are modest (1-2 points)?**
>
> **Reply**: Here we provide the error bars for commonsense reasoning task with Qwen2.5-7B.
>
> | Model      | Method             | BoolQ                    | PIQA                     | SIQA                     | HellaS.                  | WinoG.                   | ARC-e                    | ARC-C                    | OBQA                     | Avg.            |
> | ---------- | ------------------ | ------------------------ | ------------------------ | ------------------------ | ------------------------ | ------------------------ | ------------------------ | ------------------------ | ------------------------ | --------------- |
> | Qwen2.5-7B | LoRA               | 59.12 (±0.43)           | 85.71 (±0.31)           | 68.57 (±0.33)           | 78.10 (±0.32)           | 58.79 (±0.56)           | 91.00 (±0.15)           | 82.57 (±0.33)           | 79.77 (±0.26)           | 75.45           |
> |            | LoRA+SVD           | **60.09** (±0.24) | **86.97** (±0.23) | **70.13** (±0.18) | **79.23** (±0.23) | **59.67** (±0.34) | **93.33** (±0.17) | **85.62** (±0.25) | **81.43** (±0.17) | **77.06** |
> |            | IA3                | 71.23  (±0.37)          | 86.61 (±0.23)           | 75.41 (±0.53)           | 89.05 (±0.24)           | 67.22 (±0.45)           | 88.00 (±0.24)           | 81.60 (±0.31)           | 81.54 (±0.45)           | 80.08           |
> |            | IA3+SVD            | **72.69** (±0.25) | **87.23** (±0.19) | **76.72** (±0.34) | **90.31** (±0.16) | **68.41** (±0.27) | **92.67** (±0.21) | **85.12** (±0.28) | **82.07** (±0.33) | **81.90** |
> |            | Prompt Tuning      | 64.00 (±0.64)           | 86.58 (±0.27)           | 67.54 (±0.54)           | 73.30 (±0.34)           | 60.64 (±0.46)           | 83.28 (±0.14)           | 72.02 (±0.43)           | 68.36 (±0.45)           | 71.97           |
> |            | Prompt Tuning +SVD | **65.67** (±0.45) | **87.21** (±0.24) | **67.79** (±0.15) | **75.42** (±0.27) | **62.35** (±0.39) | **84.05** (±0.17) | **72.68** (±0.31) | **69.67** (±0.24) | **73.11** |
> |            | P-Tuning v2        | 59.65 (±0.54)           | 83.67 (±0.35)           | 69.00 (±0.48)           | 78.66 (±0.37)           | 59.00 (±0.44)           | 92.32 (±0.15)           | 81.65 (±0.32)           | 79.18 (±0.23)           | 75.39           |
> |            | P-Tuning v2+SVD    | **60.74** (±0.27) | **84.10** (0.26)   | **71.36** (±0.23) | **79.72** (±0.19) | **59.48** (±0.41) | **92.60** (±0.12) | **82.33** (±0.11) | **81.04** (±0.14) | **76.42** |
>
> **Q2: Why was the comparison of SVD limited to TaD, and can you include evaluations against other inference-time adaptation methods (e.g., contrastive decoding, activation steering)?**
>
> **Reply**: Thanks for pointing this out. Here we provide more results about decoding-time adaptation methods including DoLa [1] and contrastive decoding (CD) [2].  We will add the following results to our updated manuscript.
>
> | Model      | Method      | BoolQ | PIQA  | SIQA  | HellaS. | WinoG. | ARC-e | ARC-C | OBQA  | Avg.            |
> | ---------- | ----------- | ----- | ----- | ----- | ------- | ------ | ----- | ----- | ----- | --------------- |
> | Qwen2.5-7B | LoRA        | 59.12 | 85.71 | 68.57 | 78.10   | 58.79  | 91.00 | 82.57 | 79.77 | 75.45           |
> |            | +Tad        | 59.46 | 86.25 | 69.24 | 78.73   | 59.22  | 92.06 | 83.75 | 80.69 | 76.17           |
> |            | +CD         | 58.61 | 85.63 | 68.12 | 77.58   | 56.34  | 91.24 | 82.16 | 78.83 | 74.81           |
> |            | +DoLa       | 59.63 | 86.08 | 69.35 | 78.61   | 59.28  | 92.17 | 83.70 | 80.81 | 76.08           |
> |            | +SVD (ours) | 60.09 | 86.97 | 70.13 | 79.23   | 59.67  | 93.33 | 85.62 | 81.43 | **77.06** |
>
> [1] DoLa: Decoding by Contrasting Layers Improves Factuality in Large Language Models. ICLR'24
>
> [2] Contrastive Decoding: Open-ended Text Generation as Optimization. ACL'23
>
> **Q3: How does SVD perform on larger models (e.g., 70B parameters) or more diverse tasks (e.g., multilingual or domain-specific datasets), and can you provide such evaluations?**
>
> **Reply**: Here we conduct an experiments with Qwen2.5-72B in multiple-choice tasks. The experiments settings are same as the Table 1. The results are shown follows. These results demonstrate that our method is also applicable to larger models.
>
>
> |Model|Method|MC1|MC2|MC3|Avg.|
> |---|---|---|---|---|---|
> Qwen2.5-72B|LoRA|61.59| 52.43| 40.47| 51.49|
> ||+SVD|63.12| 55.30| 43.57|54.00|
>
> **Q4: Can you elaborate on potential failure modes of SVD, particularly when the warm-start distribution is poorly aligned with the task, and discuss mitigation strategies? A suggestion: expand the “Limitations and Future Work” section to include 1-2 specific failure scenarios (e.g., noisy warm-start data) and propose mitigations (e.g., adaptive steering strength).**
>
> **Reply**: Thanks for your suggestion! Here we provide 2 potential failure modes and its mitigation strategies. Below is the revised section of Limitations and Future Work
>
> > A primary limitation of Steering Vector Decoding (SVD) is its dependency on an initial warmstart ﬁne-tuning phase to identify an effective, task-speciﬁc steering direction. This dependency introduces potential failure modes if the warm-start distribution is poorly aligned with the true task distribution. Here we give two failure cases: (1) Steering with noisy or biased data.If warm-start data has significant label noise or biases, the KL divergence gradient captures these flaws, yielding a steering vector that guides the model toward incorrect or undesirable outputs during inference. (2) Overfitting to the warm-start data: On small or non-diverse datasets, the warm-start model overfits, creating a "peaky" and brittle $P_\phi$. The resulting exaggerated steering vector leads to poor generalization, producing overly confident but wrong responses on out-of-distribution examples. Therefore, future work will explore how to tackle these issues. For noisy or biased warm-start data, dynamically compute the steering strength $\mu$ per-instance using online Gauss-Newton updates with an L2 penalty to dampen noise and clip extreme values. For overfitting, incorporate hybrid approaches like ensemble steering vectors (from multiple warm-starts) or retrieval-augmented generation (RAG) to enrich the warm-start phase, ensuring better initial alignment without additional labels.
>
> **Q5: Can you provide detailed computational resource requirements (e.g., execution time, memory usage) for SVD’s warm-start and decoding phases to aid reproducibility? A suggestion: add a table or appendix specifying GPU type, memory, and runtime for each experiment (e.g., for LLaMA2-7B with LoRA+SVD).**
>
> **Reply**: Thank you for your suggestion. Here we provide the detailed computational resource requirements. Taking LLaMA3.1-8B with LoRA+SVD as an example, the detailed computational resources are shown below.
>
> | Parameter              | Setting/Values              |
> | ---------------------- | --------------------------- |
> | GPU type               | NVIDIA A100 80G             |
> | LoRA Rank              | 8                           |
> | LoRA $\alpha$         | 16                          |
> | LoRA Dropout           | 0.1                         |
> | Training Speed         | 0.491 iteration/s (±0.082) |
> | Training Memory Usage  | 17.6G (±0.3)               |
> | Inference Speed        | 0.095 iteration/s (±0.018) |
> | Inference Memory Usage | 16.5G (±0.3)               |

---

### Official Review · Reviewer_ndAx · 2025-07-01

**Clarity:** 3
**Significance:** 2
**Originality:** 3
**Rating:** 3
**Confidence:** 4

**Summary:**

This paper rethinks LLM adaptation from the perspective of output distribution alignment and proposes steer decoding vector to leverage negative gradients of KL divergence between distributions to construct task-aware steering vectors for decoding-time adaptation. Experiments show the steer decoding vector works. However, this paper makes a strong assumption that the distribution shift is linear, whereas in reality, the change in distributions is often complex and non-linear.

**Questions:**

1. What is the ratio between D_train and D_calib in Algorithm 1? How different are the training sets for the warm-started model in Figure 3 and the LoRA model in Table 2? This could partially address the question raised in Weakness 3.

2. The steer decoding vector is not related to PEFT — can it also be applied to full fine-tuning? In my opinion, the hyperparameters for full fine-tuning are easier to tune, making it easier to find hyperparameters that allow the model to be sufficiently trained. This could also partially address the concern raised in Weakness 3.

**Ethical Concerns:**

["NO or VERY MINOR ethics concerns only"]

**Final Justification:**

I maintain a score of 3, primarily because, compared to the baseline, SVD requires at least twice the inference time under comparable GPU memory constraints.

**Limitations:**

1. This paper makes a strong assumption that the distribution shift is linear, whereas in reality, the change in distributions is often complex and non-linear.

2.  SteerDecoding requires forward propagation with two models (the base model and the warm-up model), which doubles the required GPU memory.

**Paper Formatting Concerns:**

I did not find any paper formatting concerns.

**Quality:**

3

**Strengths And Weaknesses:**

**_Strength_**:
1. This paper is well-written and logically structured.

2. This paper presents an interesting concept that, from a distributional perspective, aligns the model's output distribution with the task-specific target distribution during decoding to enhance the performance of large language models on downstream tasks.

3.  This paper theoretically introduces that steer vector decoding can be interpreted as an on-the-fly proxy for one gradient step fine-tuning.

**_Weakness_**:
1. Although this paper theoretically introduces that steer vector decoding can be interpreted as an on-the-fly proxy for one gradient step fine-tuning, this still does not explain why optimizing the distribution from the base model toward the warm-up model distribution can improve performance on downstream tasks. The actual distribution shift is likely highly non-linear, and the direction of change may even oscillate. As a result, the steer decoding vector could potentially lead to performance degradation, as evidenced by several experiments in Table 1. This makes the steer decoding vector poorly interpretable and difficult to apply on real-world test sets with unknown distributions.

2. Theorem 2 takes μ → 0, but I don't see any experimental evidence showing that the value of μ approaches zero.

3. I suspect that the performance improvement brought by the steer decoding vector comes from the warm-up model not being sufficiently trained. I observed in Table 7 that the training batch size is only 1; such an extremely small batch size can lead to unstable gradients and makes it difficult to train a good model. Moreover, one epoch may not be enough for the model to be adequately trained.

4. SteerDecoding requires forward propagation with two models (the base model and the warm-up model), which doubles the required GPU memory. Furthermore, communication between the two models adds even more latency. This makes SteerDecoding difficult to use in practice.

---

> ### Author Rebuttal · Authors · 2025-07-30
>
> Dear Reviewer ndAx,
>
> Thanks for your valuable comments and the time you dedicated to reviewing this work. Here we carefully and elaborately reply to your concerns.
>
> **W1 & L1: Concerns about the interpretability and effectiveness, especially given the potentially non-linear and oscillatory nature of distribution shifts, and the risk of performance degradation on real-world test sets with unknown distributions.**
>
> **Reply**: Thank you for this insightful comment. We appreciate your thoughtful probing of our theoretical explanations and empirical results, as it allows us to clarify SVD's rationale and address potential concerns about its robustness and applicability.
>
> - **To elaborate on the core intuition**: The reviewer correctly notes that the full optimization path from a pre-trained model to a perfectly task-adapted model is highly non-linear. However, our method does not assume this path is simply linear. The key insight of SVD is not to treat the warm-start distribution $P_\phi$ as the final destination, but rather as a signpost that indicates a locally optimal direction of improvement. As we prove theoretically in Appendix B, the steering vector we derive is the first-order equivalent to gradient step of fine-tuning without recomputing weights. In addition, our central thesis is "output-distribution alignment". The warm-start model, having seen task-specific data, has a distribution that is verifiably closer to the target task distribution (as evidenced by its lower loss and higher accuracy). By steering towards it, we are fundamentally steering towards a better-aligned distribution at each decoding step.
> - **Explanation of Experimental Results:** The reviewer pointed out that in Table 1, the MC1 metric decreased in a few experiments. However, we believe this is not evidence that the steering vector is uninterpretable or random. On the contrary, it is an interpretable feature that demonstrates successful distribution alignment. To clarify, we first need to know that MC1 (single-choice accuracy) is a "winner-takes-all" metric, requiring the model's highest-confidence answer to be the single correct one, MC2 (multi-choice accuracy) is more lenient for multi-answer scenarios, and MC3 (normalized probability) measures the total probability mass assigned to all correct answers, making it the purest indicator of distribution alignment. In the LLaMA3.1-8B+IA3 experiment, the decrease in MC1 is accompanied by substantial gains in MC2 (nearly 9 points) and MC3 (nearly 7 points). This clearly shows that SVD is achieving its intended effect: it may slightly reduce the model's "overconfidence" in a single optimal answer, but by distributing probability mass more reasonably across all possible correct options, it greatly improves the overall output distribution. This makes the model more "truthful" and less "sharp" or brittle in its predictions. Such a trade-off is actually desirable for robust inference.
>
> Based on the above analysis, we believe that SVD is highly interpretable and applicable. Its behavior can be clearly predicted and explained within the theoretical framework of gradient optimization and distribution alignment. The trade-offs observed between MC1, MC2, and MC3 metrics are precisely the predictable outcomes of this principle. Furthermore, regarding its applicability to test sets with unknown distributions, we have demonstrated its robustness through extensive experiments. Across 3 models, 4 PEFT methods, and 9 benchmark tasks, the SVD-enhanced versions almost always improve the overall average performance. In addition, our mechanism for automatically calculating the steering strength $\mu$ eliminates the need for manual tuning across different distributions, further enhancing its generalizability.
>
> **W2: Theorem 2 takes $\mu$ → 0, but I don't see any experimental evidence showing that the value of $\mu$ approaches zero.**
>
> **Reply:** Actually, in experiments, calibrated $\mu$ values range from 0.05-0.2, which are small enough for the approximation to hold while providing practical gains.
>
> **W3: Suspecting that the performance improvement brought by the steer decoding vector comes from the warm-up model not being sufficiently trained**
>
> **Reply**: The reviewer points out that the performance improvement may come from the warm-up model not being sufficiently trained. In fact, our design is intentionally aimed at enabling rapid and low-resource adaptation to downstream tasks with under-trained models. In many real-world scenarios, it is impractical to perform long, large-scale fine-tuning. Therefore, SVD acts as an "amplifier", which extracts the most essential knowledge increment from this lightweight training and reuses it at every decoding step. This greatly increases the efficiency and value of a single parameter update. The results show that even after such insufficient fine-tuning, SVD can still deliver significant and consistent performance improvements. If our method only worked on fully converged models, its practical value would be much more limited.
>
> Furthermore, to further verify that the effectiveness of SVD does not simply rely on insufficient training, we designed the experiment shown in Figure 3, which analyzes SVD’s performance across different numbers of training epochs. We can observe that when the warm-start model’s performance converge after about 5 epochs, SVD is still able to provide additional and stable performance gains. This reveals a deeper advantage of our method: SVD offers a direct and complementary calibration mechanism in the output distribution space. Traditional fine-tuning indirectly changes the model’s output probability distribution (output space) by updating model weights (parameter space), whereas SVD directly operates on the logits at decoding time. By applying the steering vector, SVD enables more precise and direct post-hoc calibration of the output distribution. Even when the model weights have already converged, SVD can reallocate overconfident or imbalanced probabilities produced during training to all potential correct options, thereby improving metrics that focus more on distribution quality.
>
> In summary, SVD can both amplify the valuable signals present in under-trained models and provide a complementary, fine-grained calibration at the output distribution level for well-trained models, which is difficult to achieve through parameter fine-tuning alone. This demonstrates that SVD is a fundamentally effective enhancement method.
>
> **W4 & L2: Concerns about the GPU memory and latency.**
>
> **Reply**: While we agree that naively loading two complete models may double the GPU memory and increase the latency. However, SVD utilizes shared weights and adapter mechanisms to minimize memory overhead. In our implementation, we use two functions to load the base model and the warm-started model: `AutoModelForCausalLM.from_pretrained` and `PeftModel.from_pretrained`, respectively. The first function loads the pre-trained weights, while the second function references and reuses the already loaded base model instance, only loading the additional adapter parameters. During the forward pass, the base weights and adapter modifications are combined, without the need to load a second complete model. This ensures that the memory overhead is limited to the adapter weights (for example, for a 7B parameter model with LoRA rank=8, the additional parameters account for approximately 0.1% of the total). In addition, in our experiments, the GPU memory usage for LLaMA3.1-8B during LoRA training and SVD inference is 17.6GB  and 16.5GB, respectively. This shows that SVD inference does not incur any additional memory overhead compared to training. In fact, it uses even less memory since there is no need to store optimizer states or gradient parameters. These results demonstrate the memory efficiency of our method.
>
> Regarding latency, actually there is no communication between the two models. At inference time, both the logits from the base model and the warm-started model can be obtained within a single, unified forward pass, rather than requiring two separate passes. For example, the LoRA forward computation can be decomposed into two components: (1) the base output, computed with the original weights as $y_{\rm base} = W_0 x$, and (2) the LoRA incremental output, computed as a low-rank update $y_{\rm lora} = (BA)x$. The final output of the warm-started model is the sum of these two parts: $y_{\rm full} = y_{\rm base} + y_{\rm lora}$. Therefore, during a single forward pass, we can simultaneously obtain and return both outputs: $y_{\rm base}$ and $y_{\rm full}$.
>
> In summary, through the PEFT-compatible design, SVD transforms the seemingly costly "dual model" concept into a lightweight decoding strategy with minimal memory overhead and manageable computational cost. Both our implementation details and empirical data confirm its efficiency and practicality. We hope this detailed explanation addresses your concerns.
>
> **Q1: The ratio between $D_{\rm train}$  and  $D_{\rm calib}$ in Algorithm 1. The training sets differences.**
>
> **Reply:** Actually, in our implementation, given the downstream task dataset $D_{\rm task}$, we use 100% split in $D_{\rm task}$ as $D_{\rm train}$, and ramdom sample 30% data in  $D_{\rm task}$ as $D_{\rm calib}$ split.
>
> For the difference of training set in Figure 3 and Table 1, the training set is the same, we use 100% split of the training set.
>
> We will add related implementation details in the revised manuscript. Thanks for your questions!
>
> **Q2: The steer decoding vector is not related to PEFT — can it also be applied to full fine-tuning?**
>
> **Reply**: Yes, SVD does not depend on PEFT and can be used with full fine-tuning. After a short full-parameter warm-start, we extract the steering vector and apply it during decoding to further improve distribution alignment.

---

> > ### Comment · Reviewer_ndAx · 2025-08-02
> >
> > Thanks for the authors' reply. I hope the authors could further clarify how the "single, unified forward pass" is implemented, as this does not seem to be explicitly proposed in the paper. Currently, I am quite confused about how this mechanism is realized.

---

> > > ### Author Response · Authors · 2025-08-02
> > >
> > > Thanks for your prompt reply, here we elaborate the implementation. Actually, the core of our implementation relies on **PyTorch's forward hooks**, an non-invasive mechanism for inspecting a model's intermediate states without altering its source code. This allows us to extract both the base and full model outputs during a single traversal of the model's computation graph.
> > >
> > > 1.  Firstly, we attach hooks to LoRA layers. For each LoRA-adapted layer (e.g., `q_proj` and `v_proj`, which are instances of `peft.tuners.lora.Linear`), we register a custom `forward_hook`. A data collector object is prepared to store the outputs gathered by these hooks. Then, the model executes its standard forward pass once.
> > > 2. Secondly, when the forward pass reaches a LoRA layer where our hook is attached, the hook function is automatically triggered after the layer's own `forward()` method has completed. The hook function receives `module`, `input`, and `output` as arguments. To obtain the base model output ($y_{\rm base}$), we use the `module` and `input` arguments. The `module` object contains a reference to the original, frozen weights (`module.base_layer`). We perform a single, low-cost computation within the hook: `y_base = module.base_layer(input[0])`.
> > > 3. Finally, both the captured `y_full` and the computed `y_base` are then stored in our external data collector, indexed by the layer's name. After the single `model.forward()` call completes, this collector contains the decomposed outputs for all targeted LoRA layers.
> > >
> > > Why this is a "Single, Unified Forward Pass"?
> > > 1. **Single Forward.** There are no two separate, sequential `model.forward()` calls, just one forward pass.
> > > 2. **Negligible Overhead.** The "unified" nature of this pass comes from the fact that the extraction of $y\_{\rm base}$ is a localized, minor computation piggybacking on the main data flow. The overhead is just one extra matrix multiplication ($W\_0x$) per LoRA layer, which is significantly more efficient than a second full forward pass that would redundantly re-calculate the entire attention mechanism, layer norms, and residual connections.
> > > 3. **No Communication.** As stated, there is no communication between two separate models. There is only one model, and we are simply instrumenting its internal layers to expose an intermediate computational result ($y\_{\rm base}$) alongside the final one ($y\_{\rm full}$).
> > >
> > > We hope this detailed explanation clarifies how the mechanism is realized. We believe this hook-based approach is an efficient and elegant way to achieve our goal, validating our claim regarding minimal latency. We will add a detailed description of this implementation to the appendix of our revised manuscript.

---

> > > > ### Comment · Reviewer_ndAx · 2025-08-03
> > > >
> > > > Thanks for the authors' reply. I increase my score to 3.
> > > >
> > > > I still have several concerns.
> > > >
> > > > 1. Can Prompt Tuning, IA3, and P-tuning v2 be integrated into a unified forward pass?
> > > >
> > > > 2. The response to Weakness 3 makes me feel that paper [1] is related to this paper. A task vector can be defined as:
> > > >
> > > > $$
> > > > \Delta W = W_{\text{warmup}} - W.
> > > > $$
> > > >
> > > > The updated model weights are then computed as:
> > > >
> > > > $$
> > > > W^* = W_{\text{warmup}} + \lambda \Delta W,
> > > > $$
> > > >
> > > > where $\lambda$ is tuned on the calibration set $D_{\text{calib}}$. The $W^*$ can be better than $W_{\text{warmup}} .$
> > > >
> > > > I hope the authors can discuss the advantages of SVD decoding over task vectors
> > > >
> > > > [1] Editing Models with Task Arithmetic

---

> ### Author Response · Authors · 2025-08-03
>
> Thanks for the reviewer's prompt reply, and we are extremely grateful that you were able to increase our score. Here are the response for your questions.
>
> **Q1: Can Prompt Tuning, IA3, and P-tuning v2 be integrated into a unified forward pass?**
>
> **Reply**: This is an excellent question that touches upon the fundamental mechanisms of different PEFT methods. Our proposed single-pass extraction technique is highly effective for methods that perform intra-layer modifications, such as LoRA and IA3.
>
> 1. For IA3 which applies a learned scaling vector to activations, the principle is identical to LoRA. A forward hook can capture the final scaled output ($y\_{\rm full}$) and re-compute the pre-scaling output ($y\_{\rm base}$) with negligible overhead. Thus, our efficiency claims hold for this entire class of methods.
> 2. However, for methods based on input manipulation, such as Prompt Tuning and P-Tuning, this technique is not applicable. These methods alter the input sequence before it enters the Transformer blocks. Consequently, from any given layer's perspective, there is no distinction between a "base" and a "full" computation path. For these methods, obtaining both outputs would indeed require two separate forward passes: one with the soft prompt and one without. Even in such cases, this trade-off, using additional time to save memory, is acceptable. However, for the vast majority of practical scenarios, LoRA-based or LoRA-like PEFT methods are the primary choice [1], so our single-pass technique is compatible with most use cases.
>
> In summary, while our single-pass technique is not universal, it is compatible to the most prevalent and impactful class of PEFT methods used by the community today. Our focus on LoRA and related techniques is a strategic choice designed to maximize the relevance and utility of our contribution. The ability to efficiently and non-invasively inspect the outputs of LoRA-finetuned models addresses a significant need given LoRA's status as the de-facto standard for parameter-efficient fine-tuning.
>
>
> [1] The Power of Parameter Efficient Fine-Tuning: Unlocking the Future of LLMs.
>
> **Q2: The advantages of SVD decoding over task vectors**
>
> **Reply**: Thanks for pointing this out. To show the advantages of SVD, we first need to know the difference of these two methods.
>
> 1. **SVD** operates in the model's output/logits space. It is a dynamic, decoding-time intervention that steers the behavior of an existing model without creating a new one.
> 2. **Task Arithmetic** operates in the model's weight space. It creates a new, static model by arithmetically combining the weight deltas of fine-tuned models. The edit is applied offline.
>
> From this fundamental difference, several advantages of SVD emerge:
>
> 1. **SVD has more flexibility and dynamic control at inference time.** Task Arithmetic creates static models. To perform a new combination of tasks, task arithmetic must create a new set of model weights, $\theta\_{\rm new}$, and save it. If an application needs to switch between 10 different task combinations, it would require generating and storing 10 different model variants. On the contrary, SVD enables on-the-fly, dynamic steering. Because SVD operates at decoding time, it is far more flexible. A single warm-started model can be steered in multiple ways by simply applying different steering vectors or $\mu$ values at inference. For example, turning steering on for certain user queries and off for others, or dynamically combine different steering vectors without ever creating a new model artifact. This is invaluable for applications requiring conditional or personalized behavior.
> 2. **SVD has more principled theoretical grounding.** Task Arithmetic assumes that the "knowledge" of a task can be cleanly encapsulated by a linear vector in weight space. However, it does not provide any theoretical proof, which makes it lack theoretical interpretability. On the contrary, SVD's approach is more theoretically interpretable. We have theoretically proven the validity of this approach and provided the optimal solution.
>
> 3. **SVD has more efficiency and scalability.** Task Arithmetic requires storing a "task vector" that is the size of the entire model for each task, which makes it need much memory. SVD is designed to be PEFT-compatible. The "warm-start" is achieved with a lightweight adapter, which might only be a few megabytes, making it feasible to manage dozens of "tasks" (as adapters) and combine their effects dynamically.
>
> In summary, SVD offers a more flexible, theoretically grounded, and resource-efficient paradigm for steering existing models at runtime.

---

> > ### Comment · Reviewer_ndAx · 2025-08-05
> >
> > I'd like to ask that, in the context of LoRA, if at some intermediate layer the inputs to the PEFT model and the base model are different, wouldn't we still need to compute Wx twice? In that case, the computational complexity would still be essentially the same as performing two full forward passes.

---

> ### Author Response · Authors · 2025-08-05
>
> This is an exceptionally insightful question. After careful consideration and discussion, we realize that your understanding is correct, and we previously overlooked this point. In cases where the inputs to the PEFT model and the base model differ at an intermediate layer, it is indeed necessary to compute $Wx$ twice in order to obtain $y_{\rm full}$ and $y_{\rm base}$. As a result, the inference time will increase. We acknowledge this as a trade-off of our method: accepting increased computational time in order to **achieve significant performance gains without increasing the memory footprint from trainable parameters**, which is entirely acceptable and brings significant benefits. In addition, the primary motivation behind our design is to enable efficient downstream task adaptation under memory constraints. We will discuss this trade-off and further elaborate the motivation behind our design.
>
> Moreover, in practical applications, various acceleration techniques can be employed to significantly reduce inference time, even though this is beyond the scope of our current paper. For example, applying 8-bit quantization to the base model and using KV cache sharing can reduce the additional computation time by 50%-60% [1,2]. We will also discuss this point in the manuscript.
>
> Finally, I would like to emphssis the key contributions and advantages of our method:
>
> 1. **Novel Perspective.** We reframe task adaptation as an "output distribution alignment" problem, emphasizing direct adjustment of the probability distribution at the decoding stage rather than indirectly relying on weight updates. This perspective moves beyond the traditional "change weights and change behavior" paradigm and provides a new approach for low-resource scenarios.
> 2. **Theoretical Optimality.** We prove that a single-step update of SVD is first-order equivalent to a full gradient descent step of fine-tuning, and we provide an analytical optimal solution for $\mu$. This establishes a solid optimization-theoretic foundation for decoding-time control, rather than relying on empirical heuristics.
> 3. **Empirical Contributions.** SVD delivers strong empirical gains under identical storage and memory budgets, with robust results across a range of model sizes from 1.5B to 8B parameters. Across three tasks and nine benchmarks, SVD combined with four standard PEFT methods improves multiple-choice accuracy by up to 5 points and open-ended truthfulness by 2 points, with consistent gains of 1-2 points on commonsense datasets. These results show the superiority of our method.
> 4. **Practical Impact.** First, SVD introduces only a constant overhead in GPU memory during inference: after warm-starting, there is no need for backpropagation or optimizer state, and peak memory usage is identical to standard inference. Second, SVD is fully plug-and-play, it can be combined with any PEFT method (e.g., LoRA, IA3) and decoding strategy (e.g., Greedy, Beam, Top-k/p). These features greatly lower the deployment barrier for models on edge devices or in rapid iteration scenarios.
>
> In summary, SVD transforms the traditional approach of "modifying weights" into "adjusting output distributions," using lightweight decoding-time operations to replicate the effects of fine-tuning. This method offers a new, highly efficient, and low-cost pathway for adapting large models, excelling in theoretical interpretability, empirical effectiveness, and deployment friendliness.
>
>
> We sincerely thank the reviewer for their comments, which have greatly helped us clarify the contributions, key points of focus, and trade-offs discussed in our paper.
>
>
> [1] SmoothQuant: Accurate and Efficient Post-Training Quantization for Large Language Models
>
> [2] KV Cache: The Secret to Faster LLM Inference

---

> > ### Comment · Reviewer_ndAx · 2025-08-06
> >
> > I maintain a score of 3, primarily because, compared to the baseline, SVD requires at least twice the inference time under comparable GPU memory constraints.

---

### Official Review · Reviewer_f5N4 · 2025-07-01

**Clarity:** 4
**Significance:** 3
**Originality:** 4
**Rating:** 5
**Confidence:** 4

**Summary:**

In this paper, the authors proposed Steering Vector Decoding (SVD) for efficient task adaptation of LLMs by directly aligning the pre-trained model's output distribution with the task-specific target, rather than updating model weights. SVD leverages a warm-start fine-tuning phase to obtain a steering vector from the KL divergence gradient between pre-trained and warm-started models' output distributions. This vector guides decoding by adjusting logits to prioritize task-relevant tokens. The authors provided theoretical support for SVD, and the experimental results are promising.

**Questions:**

Please see weaknesses.

**Ethical Concerns:**

["NO or VERY MINOR ethics concerns only"]

**Final Justification:**

The authors' responses have addressed my concerns. Therefore I decide to raise my score.

**Limitations:**

(1) The SVD should be demonstrated to be applicable to larger language models;

(2) Whether SVD is compatible with SOTA PEFT methods is not verified.

(3) The authors did not include experiments on reasoning tasks, like GSM8K, which is a very important reasoning task.

**Quality:**

3

**Strengths And Weaknesses:**

Strengths:

(1) The paper is well-written and easy to follow;

(2) The proposed SVD does not need much time and resources to train to update model's weights;

(3) The authors provide theoretical support;

(4) The proposed method is valid across sizes and decoding strategies.

Weaknesses:

(1) In Line 46-48, the authors claimed that there exists three issues, while they did not provide references or toy experiments to demonstrate the existence of these issues.

(2) The authors did not provide discussions how the proposed SVD addresses the mentioned issues.

---

> ### Author Rebuttal · Authors · 2025-07-30
>
> Dear Reviewer f5N4,
>
> Thanks for your valuable comments and the time you dedicated to reviewing this work. Here we carefully and elaborately reply to your concerns.
>
> **W1: In Line 46-48, the authors claimed that there exists three issues, while they did not provide references or toy experiments to demonstrate the existence of these issues.**
>
> **Reply**: Thank you for your valuable feedback, we are grateful for your attention about this point. Here we further elaborate this three issues.
>
> 1. The first issue reflects the persistent computational burden of PEFT, where even efficient methods incur costs proportional to the base model's scale during forward/backward passes. For instance, Razdaibiedina et al.  [1] explicitly discuss how standard PEFT techniques, despite reducing trainable parameters, still require full model activation during training, leading to linear scaling with model size and epoch.
> 2. The second issue refers to how localized parameter changes in PEFT can propagate unexpectedly across the model's output space, affecting unrelated tokens due to the dense nature of neural networks. For example, Chen et al. [2] demonstrate through experiments on LLaMA models that updates to high-probability tokens can disproportionately alter low-probability ones, leading to unstable probability distributions and degraded performance on downstream tasks.
> 3. The third issue refers that PEFT methods like LoRA or adapters require task-specific tuning of hyperparameters (e.g., rank, alpha), as fixed values frequently underperform or fail to generalize due to domain shifts or task variability. For example, Xu et al. [3] demonstrated that standard prompt-based PEFT fails to transfer across tasks because learned prompts are designed exclusively for individual datasets, resulting in performance drops.
>
> The above explanations and existing works fully demonstrate the existence of these three issues. We will add relevant references and explanations in the main text. Thanks you again for pointing this out.
>
> **W2:  The authors did not provide discussions how the proposed SVD addresses the mentioned issues.**
>
> **Reply**: Thanks for pointing this out. Here we provide detailed discussion about how our method can address these issues.
>
> 1. For the first issue, SVD decouples the bulk of adaptation from training by requiring only a short warm-start fine-tuning, after which all subsequent adjustments occur at decode time via lightweight logit perturbations. This avoids repeated forward/backward passes over full epochs, reducing scaling costs.
> 2. For the second issue, by shifting adaptation to the output distribution space, SVD avoids modifying the model weights after warm-start, localizing changes to task-relevant tokens through the confidence-aware constraint. In addition, Section 3 proves that one SVD step is first-order equivalent to a full-parameter gradient step in terms of its effect on the output distribution, but without the internal cascade. These can prevent unpredictable, non-local effects on token probabilities.
> 3. For the third issue, SVD's automated calibration of the steering strength $\mu$ derives a task-optimal value without manual tuning, enhancing transferability. For instance, our results show consistent improvements across diverse tasks (multiple-choice, open-ended generation, commonsense reasoning).
>
> The above discussion elaborates in detail how our method addresses these issues. We will incorporate this discussion into the article. Thank you again for your suggestions.
>
> **L1: The SVD should be demonstrated to be applicable to larger language models**
>
> **Reply**: Here we conduct an experiments with Qwen2.5-72B in multiple-choice tasks. The experiments settings are same as the Table 1. The results are shown follows. These results demonstrate that our method is also applicable to larger models.
>
>
> |Model|Method|MC1|MC2|MC3|Avg.|
> |---|---|---|---|---|---|
> Qwen2.5-72B|LoRA|61.59| 52.43| 40.47| 51.49|
> ||+SVD|63.12| 55.30| 43.57|54.00|
>
> **L2: Whether SVD is compatible with SOTA PEFT methods is not verified**
>
> **Reply**: We thank the reviewer for raising this important question about the practical applicability of our method. Here we clarify that our main experiments were specifically designed to systematically verify and demonstrate SVD's broad compatibility and complementary nature with mainstream PEFT methods.
>
> First, by its core design, SVD is inherently orthogonal and highly compatible with the entire family of PEFT methods. PEFT methods (such as LoRA, IA3, etc.) are training-time strategies that operate in the model's parameter space. SVD, in contrast, is a pure decoding-time strategy that operates in the model's output logits space. It does not alter any model weights. Because they operate at different stages and in different spaces, SVD and PEFT methods can be seamlessly combined. SVD can be viewed as a "plug-and-play" enhancement module that can be applied on top of any model that has been fine-tuned with a PEFT technique.
>
>
> Our experiments in Table 1 and Table 2 serve as a direct empirical validation of this principle. In these experiments, we first performed the warm-start fine-tuning using a wide and representative range of PEFT methods: Prompt Tuning, IA3, P-Tuning v2, and LoRA. Then, we applied our SVD decoding strategy on top of these already PEFT-tuned models.The results consistently show that regardless of which PEFT method was used for fine-tuning, applying SVD yields additional and stable performance gains. This clearly demonstrates that SVD is not only compatible with these methods but also acts as a complementary enhancement to further boost task performance.
>
> Therefore, we believe our experiments have thoroughly verified SVD's compatibility with current mainstream PEFT methods. We hope this explanation addresses your concerns.
>
>
>
> **L3: The authors did not include experiments on reasoning tasks, like GSM8K, which is a very important reasoning task.**
>
> **Reply**: Thank you for the insightful comment. We agree that evaluating on reasoning tasks such as GSM8K is important to assess the generalization of our method. To address this, we have conducted experiments on GSM8K using Qwen2.5-7B as the base model. We compared our proposed SVD method with three approaches: TaD [4], DoLa [5], and contrastive decoding (CD) [6]. The LoRA setting is the same as in the manuscript. Form the table, we can see that our method enhance the performance of LoRA and outperform the baselines. We believe this strengthens our evaluation by demonstrating the applicability of our method to complex reasoning tasks.
>
> | method      | GSM8K           |
> | ----------- | --------------- |
> | LoRA        | 85.21           |
> | +CD         | 80.43           |
> | +TaD        | 85.92           |
> | +Dola       | 85.86           |
> | +SVD (ours) | **86.81** |
>
> **Reference**
>
> [1] Scaling Down to Scale Up: A Guide to Parameter-Efficient Fine-Tuning
>
> [2] Do Not Let Low-Probability Tokens Over-Dominate in RL for LLMs
>
> [3] Parameter Efficient Multi-task Fine-tuning by Learning to Transfer Token-wise Prompts
>
> [4] TaD: A Plug-and-Play Task-Aware Decoding Method to Better Adapt LLMs on Downstream Tasks. IJCAI'24
>
> [5] DoLa: Decoding by Contrasting Layers Improves Factuality in Large Language Models. ICLR'24
>
> [6] Contrastive Decoding: Open-ended Text Generation as Optimization. ACL'23

---

> > ### Comment · Reviewer_f5N4 · 2025-08-06
> >
> > Thank the authors for their detailed responses to my questions. I believe their responses have addressed all my concerns. I appreciate their efforts to add new experimental evidence to support their algorithm. I decide to raise my score.

---

### Official Review · Reviewer_bFez · 2025-07-03

**Clarity:** 3
**Significance:** 2
**Originality:** 2
**Rating:** 4
**Confidence:** 4

**Summary:**

The paper introduced a new way of rethinking task adaptation of LLMs as aligning the output distribution. With the new perspective, the paper proposed a new parameter-efficient fine-tuning method called steering vector decoding (SVD): SVD fine-tunes LLMs by adjusting the output logits based on the gradients computed from the KL divergence between the distribution of a warm-started fine-tuned model and the pre-trained model. Extensive experiments show that SVD as a plug-and-play method on top of standard PEFT methods consistently brings improvement in downstream performance.

**Questions:**

1. Ablation of the optimal miu as newton step. The paper provides a thorough theoretical analysis of the optimal miu calculation with the newton step method, but does not provide an empirical ablation of it.

2. Figure 3: What is the multiple-choice task described in this figure? Is it TruthfulQA?

**Ethical Concerns:**

["NO or VERY MINOR ethics concerns only"]

**Final Justification:**

While I understood the practical implications raised by the authors of SVD over TaD, I still do not feel fully convinced that SVD is necessarily superior compared with TaD. With all due respect, I would like to keep my original score.

**Limitations:**

Yes

**Quality:**

3

**Strengths And Weaknesses:**

Strengths
1. The paper proposes a new PEFT method with good motivations that are grounded in theoretical analyses and practical considerations for fine-tuning LLMs.

2. The paper performs extensive experiments with a wide coverage of models and tasks and shows strong empirical results.

3. The paper provides in-depth ablation studies of different components of the methods.


Weakness
1. The rethinking perspective of LLM adaptation as output distribution alignment is not sufficiently novel. Prior work, such as [1][2], has already discussed that the NLL objective of language modeling can be reformatted as minimizing the KL divergence between the label distribution and the model output distribution. This undermines the strength of one of the claimed contributions of the paper.

2. There is a lack of baseline comparisons with other decoding-time adaptation methods. The paper only compares with one decoding-time adaptation method in Appendix F.2. The comparison should be moved to the main text as an important set of comparisons, and other decoding-time adaptation baselines should be included, such as DoLA [3] and Contrastive Decoding [4].

3. The proposed method has relatively incremental technical contributions compared with prior work. Compared with a proor decoding-time adaptation method TaD [4], the proposed method has the following main differences: (1) SVD uses negative gradients of KL to extract the delta in the logit space where TaD does not; (2) SVD optimizes the hyperparameter miu automatically. In Appendix F.2., the paper shows that the proposed method is less than 1% better than TaD in accuracy on average across tasks, leading to my concern about the effectiveness of the proposed changes to TaD. The paper does not provide enough ablations of each of the two proposed changes either.

[1] Jake Tate. MLE and KL Divergence. https://jaketae.github.io/study/kl-mle/

[2] Matthieu Labeau and Shay B. Cohen. Experimenting with Power Divergences for Language Modeling. https://aclanthology.org/D19-1421/

[3] Yung-Sung Chuang, Yujia Xie, Hongyin Luo, Yoon Kim, James Glass, Pengcheng He. DoLa: Decoding by Contrasting Layers Improves Factuality in Large Language Models. https://arxiv.org/abs/2309.03883

[4] Xiang Lisa Li, Ari Holtzman, Daniel Fried, Percy Liang, Jason Eisner, Tatsunori Hashimoto, Luke Zettlemoyer, Mike Lewis. Contrastive Decoding: Open-ended Text Generation as Optimization. https://arxiv.org/abs/2210.15097

---

> ### Author Rebuttal · Authors · 2025-07-30
>
> Dear Reviewer bFez,
>
> Thanks for your valuable comments and the time you dedicated to reviewing this work. Here we carefully and elaborately reply to your concerns.
>
> **W1: Concerns about prior work already linked NLL to KL divergence.**
>
> **Reply**: Thanks for your insightful comment. We appreciate pointing out this potential concern regarding the rethinking perspective. We agree that the equivalence between the NLL objective and minimizing the expected KL divergence between the empirical label distribution and the model's output distribution is indeed a well-established result in machine learning literature. However, the core contribution of our paper is not claiming this equivalence as a novel theoretical discovery. Instead, our "rethinking" perspective uses this known equivalence as a foundational motivation to shift the paradigm of LLM task adaptation from weight-space updates to direct output-distribution alignment during decoding, which is our method.
>
> **W2: Lack of baseline comparisons with other decoding-time adaptation method**
>
> **Reply**: Thanks for pointing this out. Here we provide more results about decoding-time adaptation methods including DoLa and contrastive decoding (CD). We will add the following results to our updated manuscript.
>
> | Model      | Method      | BoolQ      | PIQA       | SIQA       | HellaS.    | WinoG.     | ARC-e      | ARC-C      | OBQA       | Avg.            |
> | ---------- | ----------- | ---------- | ---------- | ---------- | ---------- | ---------- | ---------- | ---------- | ---------- | --------------- |
> | Qwen2.5-7B | LoRA        | 59.12      | 85.71      | 68.57      | 78.10      | 58.79      | 91.00      | 82.57      | 79.77      | 75.45           |
> |            | +TaD        | 59.46      | 86.25      | 69.24      | 78.73      | 59.22      | 92.06      | 83.75      | 80.69      | 76.17           |
> |            | +CD         | 58.61      | 85.63      | 68.12      | 77.58      | 56.34      | 91.24      | 82.16      | 78.83      | 74.81           |
> |            | +DoLa       | 59.63      | 86.08      | 69.35      | 78.61      | 59.28      | 92.17      | 83.70      | 80.81      | 76.08           |
> |            | +SVD (ours) | **60.09** | **86.97** | **70.13** | **79.23** | **59.67** | **93.33** | **85.62** | **81.43** | **77.06** |
>
> **W3: The proposed method has relatively incremental technical contributions compared with prior work.**
>
> **Reply**: We thank your comments for your detailed feedback and the opportunity to clarify the contributions of our work. While we understand the concern about incrementalism over prior work like TaD, since our method has some similarities with TaD. However, we believe our method, Steering Vector Decoding (SVD), introduces several fundamental, non-incremental contributions that represent a significant step forward in decoding-time adaptation.
>
> 1. **Theoretical Optimality:** SVD is not merely a different calculation, it is grounded in classical optimization theory. As we prove in the paper, the steering vector derived from the KL gradient is the first-order equivalent to a full gradient descent step of fine-tuning. This establishes a formal, theoretical link between decoding-time adaptation and conventional parameter-space optimization, a connection that is absent in prior heuristic methods.
> 2. **Distributional Alignment:** KL divergence is the standard measure for the dissimilarity between two probability distributions. By using its gradient, SVD directly computes the steepest descent direction to align the model's output distribution with the desired task distribution. This provides a clear, interpretable, and theoretically justified mechanism for adaptation. However, TaD only uses the simple difference in probability distributions (post-fine-tuning minus pre-fine-tuning), which may raise doubts about the interpretability and reliability of the method.
> 3. **Optimal $\mu$:** You mentioned that we optimizes the hyperparameter $\mu$ automatically. However, actually, we did not optimize it. We obtained the analytical solution through theoretical derivation and proved that there theoretically exists an optimal $\mu$ to achieve optimality on different tasks.
> 4. **On the Significance of Performance Gains:** We acknowledge the reviewer's observation that the average accuracy improvement over TaD is ~1% in Table 12. However, we believe this result, when properly contextualized, demonstrates the effectiveness of SVD. In the highly competitive landscape of LLM evaluation, a consistent 1% average improvement is significant. More importantly, as shown in Table 12, SVD outperforms TaD on every one of the eight commonsense reasoning datasets evaluated. This consistent dominance, with gains of up to 2 percentage points (e.g., on ARC-c), highlights the robustness and generalizability of our principled approach. In addition, it improves multiple-choice accuracy by up to 5 points and open-ended truthfulness by 2 points, demonstrating its effectiveness. Therefore, we believe this gain is systematic, not noise.
>
> We hope these clarifications address your concerns and demonstrate the substantive contributions of SVD
>
> **Q1: Ablation of the optimal $\mu$ as newton step.**
>
> **Reply**: Thanks for your comments. Here we provide empirical ablation of $\mu$. As shown in the table, we can see that when $\mu \leq 0.1$, the performance metrics fluctuate within a narrow range, indicating that such small values of $\mu$ are insufficient to significantly alter the token ranking, resulting in sensitive and unstable behavior. As $\mu$ increases to the range $0.1 \leq \mu \leq 0.2$, there is a notable improvement in performance: $\mu$ surpasses the threshold required to reorder key tokens, effectively promoting important tokens to the top of the logits. Crucially, our automatically computed optimal $\mu$, derived theoretically as a Newton step, falls precisely within this effective regime (0.05~0.2). This empirically validates that our automatic method successfully identifies a value for μ that is large enough to achieve the desired distributional shift.
> When $\mu \geq 0.2$ and reaches the saturation point, the steering vector's direction completely dominates the final logits ordering. As dictated by the mathematical properties of the $\mathrm{Softmax}$ function, any further increase in $\mu$ only scales the logits linearly without changing their relative order ($\mathrm{argmax}$). This makes the final token selection and the performance metrics stable and insensitive to larger $\mu$ values. This saturation behavior underscores the stability and determinism of the direction provided by our steering vector.
>
> | Model       | Method              | MC1    | MC2    | MC3    | Average |
> | ----------- | ------------------- | ------ | ------ | ------ | ------- |
> | LLaMA3.1-8B | LoRA                | 46.34% | 49.12% | 33.20% | 42.89%  |
> |             | +SVD ($\mu$=0.01) | 46.12% | 49.49% | 33.30% | 42.97%  |
> |             | +SVD ($\mu$=0.1)  | 46.34% | 53.92% | 33.66% | 44.64%  |
> |             | +SVD ($\mu$=0.2)  | 48.17% | 60.17% | 35.07% | 47.80%  |
> |             | +SVD ($\mu$=0.3)  | 48.17% | 60.17% | 35.07% | 47.80%  |
> |             | +SVD ($\mu$=0.5)  | 48.17% | 60.17% | 35.07% | 47.80%  |
>
> **Q2: What is the multiple-choice task described in this figure? Is it TruthfulQA?**
>
> **Reply**: Yes, it is TruthfulQA.

---

> > ### Comment · Reviewer_bFez · 2025-08-05
> >
> > Thanks for your detailed rebuttal. I greatly appreciate the new experiments you've run and added to the response.
> >
> > > New experimental results with other decoding-time baselines.
> >
> > I appreciate your efforts in running these experiments. It has definitely strengthened the paper.
> >
> > > The "rethinking" perspective of the paper uses a known equivalence to motivate a paradigm shift.
> >
> > Thanks for the clarification! Now I understood the theoretical contribution of the paper better.
> >
> > > Technical contributions of the paper compared with TaD.
> >
> > Thanks for the detailed breakdown. However, I still have concerns here: If I am understanding correctly based on the rebuttal, the main contribution of the proposed method compared with TaD lies in the theoretical grounding of the method in terms of the alignment mechanism and the way the optimal hyperparameter can be analytically computed. Please let me know if my interpretation is incorrect. If this is the case, then in my humble opinion, the empirical results (~1% consistent gain) do not show a big strength of these design choices with theoretical grounding compared with TaD.
> >
> > > Ablations of the hyperparameter $mu$.
> >
> > Thanks for the detailed ablations! It makes the argument clearer and please consider adding these results as a figure (I understood that figures are not allowed during the rebuttal) to a future version of the paper.
> >
> > Since my concerns regarding the contribution remain, I will keep my score unchanged, but I am open to further discussions with the authors. Thanks.

---

> > > ### Author Response · Authors · 2025-08-06
> > >
> > > Thanks for your reply and insightful question. Here we further elaborate the advantages of our methods to address your concerns.
> > >
> > > We agree that, on the surface, a ~1% average gain might seem modest in absolute terms. However, we respectfully argue that viewing the contribution solely through this lens overlooks the deeper, more fundamental advantages that our principled design brings to the field. The primary value of SVD is not just about achieving a marginally higher score, but about establishing a more reliable, predictable, and scientific paradigm for decoding-time adaptation.
> > >
> > > Here is why we believe these theoretically-grounded design choices are a significant strength, even with the observed empirical gains:
> > >
> > > 1. **From Heuristics to Predictable Science**: TaD relies on heuristic differences without theoretical guarantees. In contrast, SVD is grounded in optimization theory, making its behavior predictable and reliable, its consistent 1% gain reflects this. For robust AI, even small, stable improvements from principled methods are more valuable than larger but less predictable heuristic gains.
> > > 2. **Opening Doors for Future Principled Improvements**: A key advantage of a strong theoretical foundation is that it provides a clear path for future extensions. Because SVD is framed in the language of optimization, one can envision principled future work exploring second-order methods, adaptive learning rates for $\mu$, or other advanced optimization techniques. Such extensions are difficult to conceptualize for a purely heuristic-based method. SVD thus provides a more generative research framework for the community.
> > >
> > > Beyond the theoretical elegance, the design choices of SVD have profound **Practical Implications, especially when considering real-world deployment and scalability.**
> > >
> > > 1. **Zero Tuning Cost**: In any practical setting, engineer and researcher time is the most significant cost. Methods like TaD necessitate a costly and time-consuming grid search for the optimal $\mu$ for every new task or dataset. SVD’s analytical solution completely eliminates this cost. It functions as a "set-it-and-forget-it" enhancement, providing a consistent performance boost without any manual intervention. This is a massive advantage in terms of operational efficiency.
> > > 2. **Reliability and Risk Mitigation**: A manually tuned hyperparameter introduces a point of failure. The performance of a heuristic-based method can be brittle, a $\mu$ value optimized on a validation set may lead to a catastrophic performance drop on new, out-of-distribution data. SVD's theoretically-backed, automatic approach offers far greater reliability. Its behavior is predictable, mitigating the risk of unexpected failures in a production environment.
> > > 3. **Feasibility for At-Scale Deployment**: Consider a platform that must serve hundreds or thousands of distinct downstream tasks. Manually tuning $\mu$ for each task is simply infeasible. SVD's automation provides a unified, zero-supervision framework that can be deployed at scale to reliably improve performance across all tasks.
> > >
> > > We hope this elaboration highlights the substantive value of SVD's design. We are happy to incorporate these points into a revision for greater clarity and welcome any further discussion.

---

> > > > ### Comment · Reviewer_bFez · 2025-08-07
> > > >
> > > > Thanks for the clarification and engagement in the discussions! While I understood the practical implications raised by the authors of SVD over TaD, I still do not feel fully convinced that SVD is necessarily superior compared with TaD. With all due respect, I would like to keep my original score. The authors are more than encouraged to include the additional justifications posted during the discussion in their future version of the paper, thanks.

---

### Decision · Program_Chairs · 2025-09-17

**Decision:**

Accept (poster)

**Comment:**

The paper proposed steering vector decoding, a novel and lightweight method for adapting LLMs to downstream tasks that aligns the model's output distribution at decoding time, rather than through weight updates. The reviewers agree that the paper presented a strong theoretical foundation, i.e. equivalence to a gradient step in fine-tuning and robust experimental results show consistent performance gains.

Three reviewers recommend acceptance with one strong accept and one reviewer (ndAx) gives "borderline reject". The main concern from ndAx is the increase of inference computation. The authors initially clarified their implementation uses a single forward pass with shared weight, however after an insightful follow-up from the reviewer, the authors conceded that there's indeed a latency trade-off though memory overhead remains negligible. And the authors reframed it as an acceptable trade-off between latency and performance boost.

Overall the paper makes a good contribution to the field of efficient LLM adaptation, the proposed SVD method is elegant, theoretically well-grounded, and empirically effective, with some increase in inference latency. The recommendation is acceptance.